# The impact of methane leakage on the role of natural gas in the European energy transition

Behrang Shirizadeh [1,2] ✉, Manuel Villavicencio [1], Sebastien Douguet [1], Johannes Trüby[1], Charbel Bou Issa [1], Gondia Sokhna Seck [3], Vincent D'herbemont [3], Emmanuel Hache [3], Louis-Marie Malbec [3], Jerome Sabathier [3], Malavika Venugopal[4], Fanny Lagrange [4], Stephanie Saunier[4], Julian Straus [5] & Gunhild A. Reigstad [5]

Decarbonising energy systems is a prevalent topic in the current literature on climate change mitigation, but the additional climate burden caused by methane emissions along the natural gas value chain is rarely discussed at the system level. Considering a two-basket greenhouse gas neutrality objective (both $CO_2$ and methane), we model cost-optimal European energy transition pathways towards 2050. Our analysis shows that adoption of best available methane abatement technologies can entail an 80% reduction in methane leakage, limiting the additional environmental burden to 8% of direct $CO_2$ emissions (vs. 35% today). We show that, while renewable energy sources are key drivers of climate neutrality, the role of natural gas strongly depends on actions to abate both associated $CO_2$ and methane emissions. Moreover, clean hydrogen (produced mainly from renewables) can replace natural gas in a substantial proportion of its end-uses, satisfying nearly a quarter of final energy demand in a climate-neutral Europe.

Fighting climate change is one of the main sustainable development goals (SDG)[1]. Limiting global warming to 1.5 °C requires global greenhouse gas (GHG) neutrality no later than 2050[2]. Decarbonisation of energy systems has recently gained significant attention in the scientific literature, with a strong consensus on the effectiveness of large-scale renewable development and efficiency improvements[3–8]. Sectoral studies focusing on electricity systems[8–11], transport sector[12,13], industries[14,15] and buildings[16,17] confirm the findings of more integrated general energy studies, highlighting that electrification and the shift towards clean hydrogen in hard-to-abate sectors are the key enablers for achieving climate neutrality, regardless of the considered geography. According to the existing literature, natural gas might still play a sustainable role in a carbon-neutral energy system, if associated with carbon sequestration processes such as carbon capture, utilisation and storage (CCUS)[18–21]. Although $CO_2$ emissions are the focus of attention of these studies, anthropogenic methane emissions along the natural gas value chain are a major contributor to global warming[22]. Their global warming potential (GWP) is 29.8 (over 100 years) to 82.5 (over 20 years) times that of $CO_2$[23].

Global Warming Potential is one of the most widely used climate metrics to assess the relative potency of different GHG emissions (such as $CH_4$), in comparison to the reference gas: $CO_2$. GWP can be estimated over a chosen time frame, 20 ($GWP_{20}$) and 100 ($GWP_{100}$) years being the most common time frames. Both metrics have evolved to be the 'default' metrics in the policy arena. Most scientific literature, assessing the impacts of greenhouse gases on climate change assess longer time effects, using $GWP_{100}$. However, Intergovernmental Panel on Climate Change (IPCC) in its last assessment report[23] highlights that the metric highly depends on the considered context and the period during which the $CO_2$ emissions should be stabilised in the atmosphere. In the context of this assessment, aiming for GHG-neutrality by 2050, $GWP_{20}$ has been given priority. From a climate change

[1]Deloitte Economic Advisory, 6 Place de La Pyramide Tour Majunga Deloitte, 92800 Puteaux, France. [2]CIRED, 45 bis avenue de La Belle Gabrielle, 94736 Nogent sur Marne Cedex, France. [3]IFP Energies Nouvelles, 1-4 Avenue Bois Preau, 92852 Rueil-Malmaison, France. [4]Carbon Limits, C. J. Hambros plass 2, 0164 Oslo, Norway. [5]SINTEF Energy Research, Sem Sælands Vei 11, 7034 Trondheim, Norway. ✉e-mail: bshirizadeh@deloitte.fr

perspective, methane emissions must also be accounted for[20] to avoid climate neutrality being delayed by two extra decades[24]. Although assessments vary, the social cost associated with methane emissions is much higher than that of their abatement; Erickson et al. estimate this social cost to be between $471/tCH_4 and $1,570/tCH_4[25], while according to International energy Agency's (IEA) methane tracker[26], methane abatement technologies are often cost-effective. Methane Guiding Principles' cost model shows an average methane abatement technology cost below $140/tCH_4[27].

This article aims at identifying cost-optimal European energy transition pathways, and more precisely, the role of natural gas and its derivative products when considering a two-basket GHG-neutrality objective, including both $CO_2$ and methane emissions by 2050. While integrated assessment models (IAMs) can generally include multiple GHGs, they lack the detailed representation of the energy system and temporal and spatial precision, they are generally based on top-down allocation of energy sources and carriers and they do not result from explicit optimisation. Therefore, we present a modelling framework with a detailed representation of the European energy system via soft-linking two different models: an energy system optimisation model (MIRET-EU) and a hydrogen import model (HyPE). Different methane abatement routes have been considered to analyse the evolution of the role of natural gas in the European net-zero paradigm. Clean hydrogen (produced from low-carbon electricity via electrolysis or natural gas with abated $CO_2$) can be considered as one of the key natural gas replacement options in different hard-to-abate sectors, but it can also be a key consumer of natural gas[5,18,28]. Therefore, our analysis is complemented with a detailed analysis of the role of clean hydrogen in a climate-neutral Europe and its supply routes. The impact of methane emissions on unit hydrogen production emissions has been studied[29–31], but not within the context of the decarbonisation of the overall energy system. This integrated approach sheds light on the synergies between a profound transformation of the energy system and the role of hydrogen and natural gas, considering methane and $CO_2$ emissions neutrality while excluding natural gas and hydrogen imports from Russia.

## Results
### Methane emissions can follow different pathways
Methane emissions along the natural gas value chain depend considerably on the upstream, midstream and downstream practices established[32]. These emissions in turn have an impact on the unit

emissions of natural gas-based hydrogen production[33]. We assessed current best understanding of emission levels and three scenarios to estimate the methane emission factor (EF) trajectory between 2019 and 2050. Emission factor is a coefficient that quantifies the emissions or removals of a gas per unit activity. Methane emission factor in this report is estimated as the sum of methane emissions along the natural gas value chain, divided by their respective activity data, such as volume of methane produced, methane transported, methane imported, etc. This was done for over 30 countries, consisting of countries exporting gas to Europe (1) via pipeline and (2) via liquified natural gas (LNG) cargoes, (3) exporting low-carbon hydrogen to Europe and (4) European gas producing and (5) gas or LNG importing countries. The EFs from different countries were combined into a weighted-average EF for natural gas consumed in Europe, depending on the share of natural gas entering from different countries. The EFs are converted to $CO_2$ equivalents ($CO_{2eq}$) in the model, using a GWP of 20 years (82.5)[23]. This allows to treat methane and $CO_2$ emissions on the same basis to assess the trade-offs and investments necessary to achieve climate neutrality by 2050 (see the Methods section for more information on the choice of the global warming period considered).

The current emissions (CEF) scenario represents the current best understanding of emissions and assumes constant emissions until 2050 in each country. The EF of natural gas consumed in Europe in 2019 is estimated to be 8.7 ktCH_4/bcm in this scenario. This accounts for 35% of the $CO_{2eq}$ (in GWP_20) arising from the combustion of natural gas in Europe. Reduction in overall EF in this scenario from 2019 to 2050 (Fig. 1) is made possible by altering the mix of natural gas entering the sub-continent. Owing to the difference in methane EFs among natural gas and/or LNG exporting countries, the origin of natural gas consumed in Europe affects the overall EF.

The harmonised pledges (HP) scenario represents the emissions based on current announced policies[34,35], Nationally Determined Contributions (NDC)[36] and methane pledges in each country[37–40]. The best available technologies (BAT) scenario assumes EFs that could be achieved by adopting BAT by a certain year. The assumed year of BAT deployment depends on the country. IEA methane abatement options and industry targets were applied to countries on a case-by-case basis. In all three scenarios the EF drops progressively until 2050. The methane EF in the BAT scenario is respectively 70% and 65% lower than the 2050 EF in the CEF and HP scenarios. The BAT scenario sees the sharpest decrease in methane's additional environmental burden by 2050. These results show that there is significant room for further methane emission reduction (BAT), more than what is envisaged under the existing policy framework (HP).

### Highly renewable future with different paths for natural gas
The European energy mix was highly fossil-dependent in 2016, with coal, oil and natural gas representing >1100 Mtoe of primary energy demand (73%). By 2030, following the phase-out of Russian gas, the share of natural gas shrinks across all scenarios (Fig. 2.a). By this date, the LNG import capacity in Europe is limited and only the BAT scenario sees an increase in LNG imports (about 40% higher than historical levels), partially replacing imports from Russia. Fossil energy sources represent <800 Mtoe of primary energy by this horizon. Although the chosen methane emission scenario has a strong influence on the role of natural gas and renewables in the energy mix in the long term (2050), oil and coal are set to dwindle: no matter the scenario, reaching net-zero by 2050 requires a near phase-out of these two energy sources.

The only difference between scenarios being the assumed trend of methane emissions of natural gas, we find that higher methane footprints lead to higher substitution of natural gas by renewable energies. Natural gas represents only 9% of primary energy demand in 2050 in the CEF scenario, where the footprint of natural gas is at the current high levels, while it represents as high as 26% of the

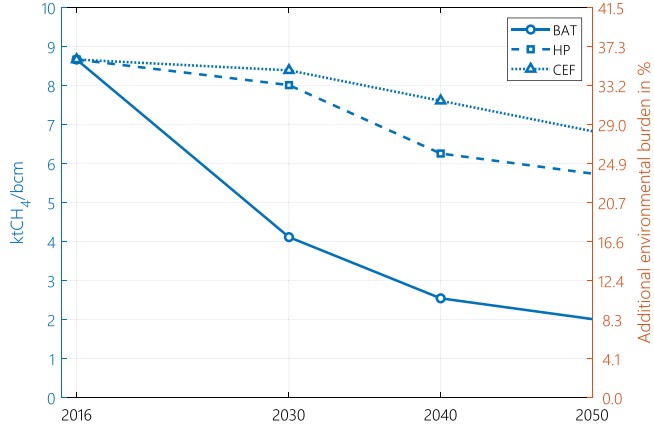

**Fig. 1 | Methane intensity of natural gas consumed in Europe for each methane emission scenario.** The "Additional environmental burden" represents the additional $CO_{2eq}$ (in GWP_20) associated with the methane footprint of a unit of natural gas consumed in Europe, in percentage of the combustion EF of natural gas. BAT stands for the best available technology scenario, HP stands for the harmonised pledges scenario and CEF stands for the current emission factors scenario.

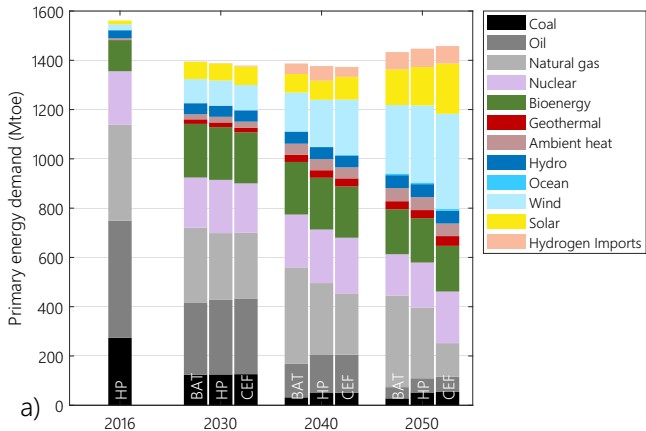

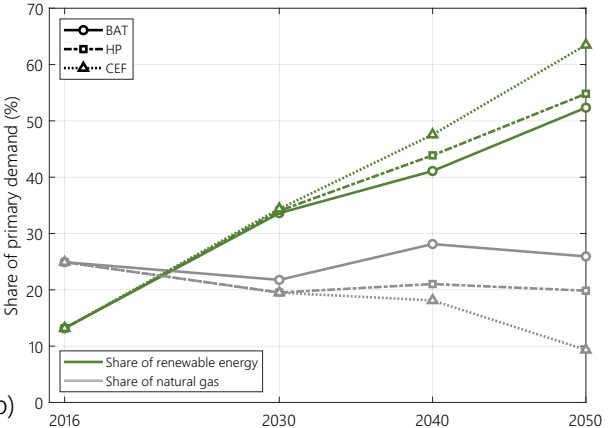

**Fig. 2 | Evolution of primary energy demand. (a)** Energy supply mix and (**b**) the share of renewables and natural has for each of the methane emission cases, compared to the historical levels (2016). BAT, HP and CEF stand for the best available technology, harmonized pledges and current emission factors scenarios respectively. Primary energy decreases by at least 100 Mtoe between 2016 and 2050. This decrease is mainly due to the massive replacement of fossil fuels by renewables, where the energy supply is already mostly in its final consumption form (for instance electricity for wind and solar power and hydroelectricity). In contrast, in a highly fossil-based energy system, the primary energy demand tends to be higher due to conversion losses of fossil energy sources to final end-uses (electricity, transport, low-temperature heating, etc.).

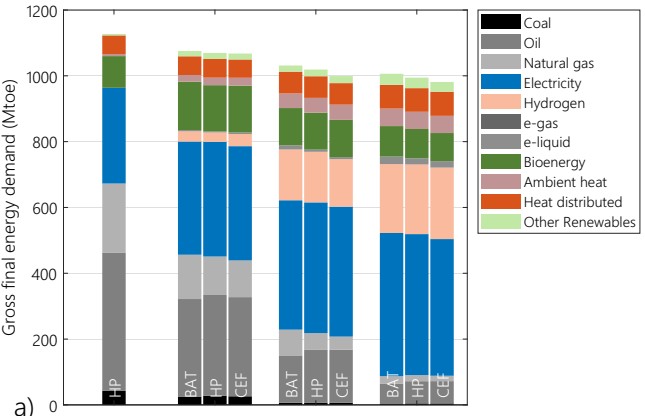

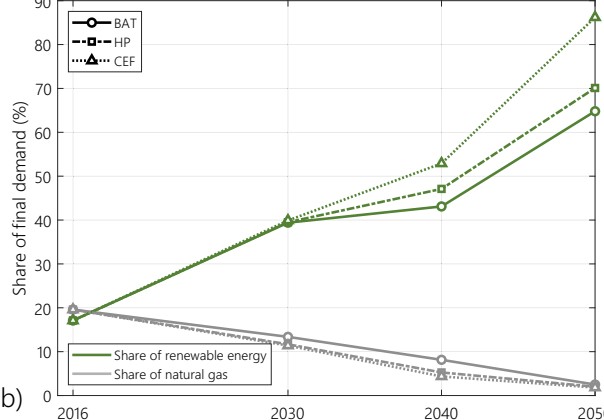

**Fig. 3 | Evolution of gross final energy consumption. (a)** Energy consumption mix by energy vector and (**b**) the share of renewables and natural has for each of the methane emission cases, compared to the historical levels (2016). BAT, HP and CEF stand for the best available technology, harmonized pledges and current emission factors scenarios respectively. Final energy demand experiences an 11% to 13% decrease between 2016 and 2050. In a growing economic environment with positive GDP (gross domestic product) growth, the growth in economic activities is expected to lead to higher final energy demand. In our analysis, while the final energy in the form it is consumed (for instance transport demand in tonne-kilometres and heating demand in the form of thermal energy demand–TWh$_{th}$) soars, the final energy carrier's demand quantity shrinks. This decrease is due to efficiency measures taken in the industrial processes by 2030, as well as the shift to more efficient final end-use energy carriers such as electricity and hydrogen. For instance, both electric vehicles and heat pumps for space heating are about two to three times more efficient than their combustion-based counterparts (internal combustion engine vehicles and boilers).

primary energy mix in the BAT scenario where methane emissions are the most abated (Fig. 2.b). Such a high share for natural gas in the primary energy mix means that about 70% of European natural gas consumption should be satisfied by LNG imports in the long run. Compared to the historical import levels, this amounts to two- (HP) to nearly four-fold (BAT) increase in European LNG imports.

Renewable energy sources represented <15% of the primary energy mix in Europe in 2016. They become the dominant source of the energy supply by no later than 2040, whatever the methane emission scenario. Their share reaches between 41% and 48% of the primary energy mix by 2040 and 52% to 63% of it by 2050 (Fig. 2.b). Correspondingly, the share of renewable energy in gross final energy consumption reaches between 65% (in the BAT scenario) and 86% (in the CEF scenario) by 2050. This increase is mainly due to the accelerated deployment of wind and solar power, experiencing an 11- to 15-fold increase in their energy supply.

The evolution of the final energy mix is hardly influenced by methane abatement, as the final energy consumption of each of the main energy vectors remain robust to the methane leakage scenario

(Fig. 3). By 2050, 415 to 435 Mtoe of final energy consumption is in the form of electricity and 209 to 217 Mtoe in the form of hydrogen, while natural gas amount to between 16 and 23 Mtoe only, out of nearly 1000 Mtoe of final energy consumption. Therefore, the share of natural gas in final energy consumption remains around 2% in all scenarios. This is due to the fact that most of natural gas use in the energy system is directed to the production of transformed energy carriers: electricity and hydrogen.

All scenarios show a radical shift in other end-use carriers: in 2016, oil held the largest share in final energy consumption (37%), while electricity represented only a quarter. As unabated uses of fossil fuels are incompatible with net-zero goals, their consumption is either reduced and limited to sectors where $CO_2$ can be abated: by 2050, the share of electricity in the final energy consumption reaches up to 43%

(above 400 Mtoe), while oil and coal together represent below 7% (between 64 Mtoe and 72 Mtoe). Hydrogen becomes the second most important end-use fuel. It represents nearly a quarter of the final energy consumption by 2050, replacing natural gas demand to a big extent.

### Each pathway needs a different emissions offset

All scenarios lead to net-zero emissions for the combined methane and $CO_2$ emissions (Fig. 4). However, the distribution between methane and $CO_2$ emissions varies, especially for 2030 and 2040. While methane emissions in 2030 follow the expected trend with the BAT scenario having the lowest methane emissions, followed by the HP and CEF scenarios, the differences in methane emissions in 2040 narrow down and methane emissions of HP exceed CEF in 2050. The change of position between the HP and CEF scenarios in 2050 is due to the significant reduction in natural gas use in the CEF scenario in 2050, corresponding to 50% of the natural gas used in the HP scenario. Yet, both the HP and CEF scenarios have significantly larger methane emissions despite the lower natural gas primary energy demand, highlighting the importance of methane emissions' abatement. Net removal of $CO_2$ from the atmosphere is required in 2050 in all scenarios to offset unabated methane emissions along the natural gas value chain. HP and CEF require 50% and 15% more net $CO_2$ removal than the BAT scenario. Removal is achieved through a combination of bioenergy with CCS (BECCS) in the power, industry and hydrogen production sectors and direct air carbon capture and storage (DACCS).

### Hydrogen is one of the key pillars of climate neutrality

Hydrogen has a key role in the shift from fossil fuels under the European net-zero target. This can be observed by the resilience of its demand level to the methane emission scenario (Fig. 5). From 2030 onwards, hydrogen becomes an essential energy carrier for the industry and the transport sectors. In the latter, hydrogen is mostly used directly in fuel cells to decarbonise heavy-duty vehicles and through the production of e-fuels for aviation (21 to 26% of its final energy demand by 2050) and bunkers with about 60% of the maritime transport final energy demand (the analysis considers only the European maritime and aerial transport and inter-continental bunkers and aviation are excluded from the analysis. It is worth mentioning that these two transport sub-sectors do not reach climate neutrality on their own by 2050, and they use the offset generated by BECCS and DACCS.). In the industry sector, hydrogen replaces fossil fuels,

particularly for iron and steel production (17 $MtH_2$ consumed in 2050). The use of hydrogen through e-fuels in the aviation mostly unfolds towards the end of the transition period (i.e., 2040-2050). In 2050, the total demand for hydrogen amounts to nearly 100 $MtH_2$.

The hydrogen production mix generally consists of a combination of renewable hydrogen and low-carbon hydrogen produced in Europe, complemented by non-European imports. The fast deployment of reformers with CCUS enables a rapid launch of the hydrogen economy in all three scenarios, with >50% of hydrogen produced with them in 2030. In the long run, renewable hydrogen based on electrolysis takes over as the primary source of hydrogen in Europe, providing between 50 $MtH_2$ (BAT scenario) and 75 $MtH_2$ (CEF scenario). By then, the contribution of natural gas reformers largely depends on the mitigation of methane emissions along the natural gas value chain: in the BAT scenario, reformer-based low-carbon hydrogen reaches 25% of the total hydrogen production by 2050, while in the CEF scenario with the highest methane EF, European hydrogen is produced almost exclusively via electrolysis. In all three scenarios European clean hydrogen demand is not fully satisfied by domestic production and is complemented by imports (up to 25% of European hydrogen demand by 2050).

### Hydrogen trade is a must

Europe becomes a net importer of hydrogen by no later than 2030, regardless of the scenario, though with limited uptake (<1 $MtH_2$). By 2050, imports start playing a significant role in complementing European hydrogen production and sustaining the ambitious development of the European hydrogen economy. They reach about 25 $MtH_2$ by 2050 (Fig. 6). Nearly 70% of this hydrogen is produced from renewable electricity through electrolysers. The main renewable hydrogen exporters to Europe are the North-African countries that benefit from significant solar irradiation and available surface for solar PV installations: Tunisia, Morocco, Algeria and Egypt. The remaining 8.7 $MtH_2$ of hydrogen imports come from reformer-based low-carbon hydrogen production from traditional natural gas exporters to Europe: Algeria, Qatar and Saudi Arabia. The main entry points of the hydrogen imports in Europe are, in descending order, Spain, Italy, France, the Netherlands and Bulgaria.

## Discussion

The methane abatement scenario analysis highlights two important results: (1) methane abatement, alongside with carbon capture and storage, as well as increased LNG imports are crucial for a continued

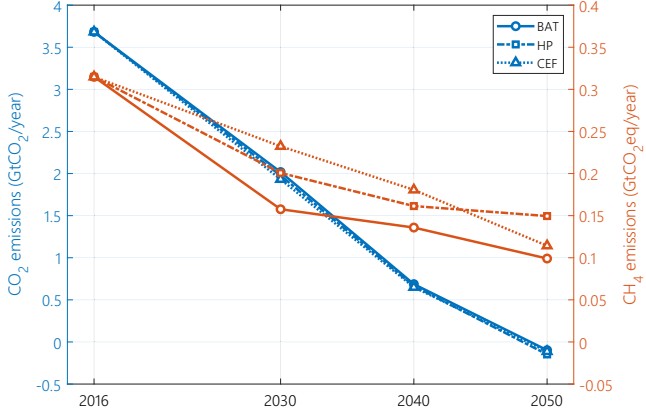

**Fig. 4 | The evolution of $CO_2$ emissions (left axis, blue) and methane emissions (right axis, orange) for the three emission scenarios from the historical values (2016) to 2050.** Both $CO_2$ and methane emissions are represented in $GtCO_{2eq}$/year terms, but the scale of the secondary vertical axis (methane emissions) is an order of magnitude smaller than the primary vertical axis. The best available technology scenario is represented by abbreviation BAT, harmonized pledges by HP and current emission factors by CEF.

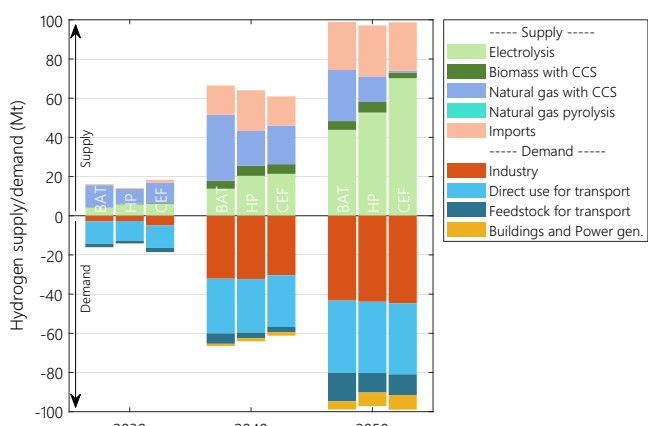

**Fig. 5 | Evolution of the hydrogen supply and demand for each of the methane emission scenarios between 2030 and 2050.** The positive values indicate hydrogen supply, while the negative values indicate hydrogen demand. BAT, HP and CEF stand for the best available technology, harmonized pledges and current emission factors scenarios respectively.

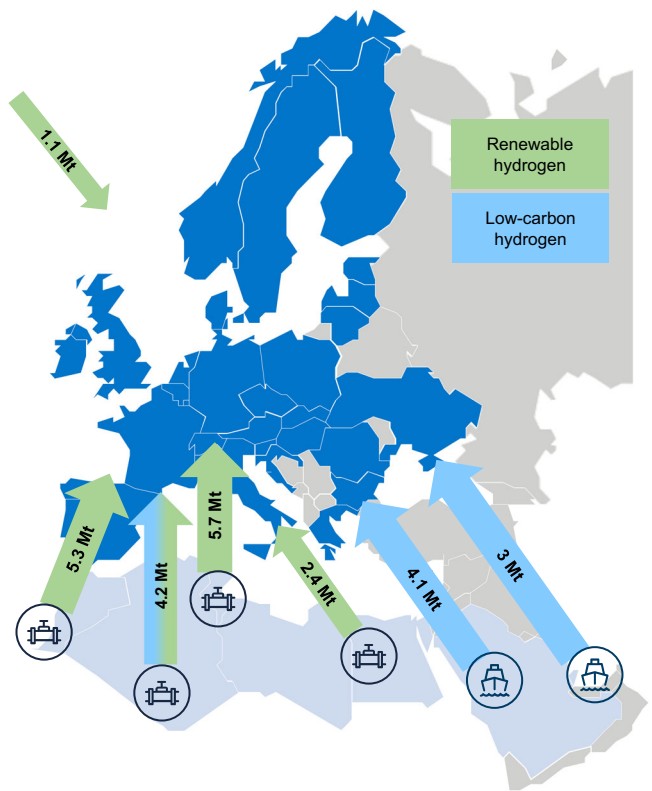

**Fig. 6 | European hydrogen imports from the neighbouring regions for the best available technology scenario.** The green colour represents renewable hydrogen (electrolysis based on wind and solar power) imports and the blue colour represents low-carbon hydrogen (reformation of natural gas with CCS) imports. Maritime hydrogen imports (via ammonia shipping) are shown with a ship logo, while pipeline gaseous hydrogen imports are shown with pipe logos.

role of natural gas with larger abatement (BAT compared to HP) resulting in a more pronounced role. In fact, (2) the current policy framework for methane emission reduction along the natural gas value chain (HP) entails very limited methane leakage reductions, ultimately degrading the role of natural gas in the climate-neutral European energy system.

The continuous role of natural gas in the future climate-neutral energy system requires significant determination both at the policy and industry sides. Our study shows that only CCS is not the sole requirement of continued role of natural gas. Even with large-scale CCS development, current methane leakage levels lead to a near phase-out of natural gas from the climate-neutral European energy mix (CEF). While historically neglected, deployment of best available methane abatement technologies is cost effective. Even with very low natural gas prices such as $20/MMBtu, a methane abatement cost of $140/tCH$_4$ accounts for no more than 1% of the natural gas price considering current emission factors. BAT rollout is also a very robust strategy for the oil and gas companies, that can guarantee a future for natural gas if combined with large-scale CCS deployment. The extra cost of these technologies remains negligible compared to natural gas production costs and carbon abatement options, while the future natural gas market in such a paradigm (BAT) can be nearly three times a future with no methane abatement (CEF). The difference between the results of BAT and CEF scenarios and their methane emissions calls for including methane leakage in GHG policies, either via inclusion in taxation or quota mechanisms. Moreover, to avoid extra-European carbon leakage, upstream methane emissions should be included in the European carbon border adjustment mechanism.

Russian natural gas imports historically accounted for nearly half of European imports[41]. Their phase-out leads to a significant increase in LNG imports, notably for the BAT and the HP scenarios where the continued role of natural gas is most pronounced. The natural gas supply levels in these scenarios remain respectively at 96% and 74% of the current levels, which implies that the removal of Russian natural gas supply is mainly met by the global LNG market[42]. This reduces the risk for security of supply by avoiding dependence on a single country for a large share of the natural gas supply. However, it also increases the risk associated with exposure to the global LNG market and its price dynamics.

Renewable energies are the key building blocks for a GHG-neutral energy system, especially for electricity and hydrogen supply. Renewable energies, electrification and clean hydrogen (mainly renewable) are no-regret solutions whose contributions are crucial, independent of the methane abatement scenario. If methane emissions associated with the natural gas value chain are not reduced, investments in both renewable energy and electrolysis capacity must however be accelerated. Notably, cumulative installed electrolyser capacity is between 46% to 64% higher in the HP and CEF scenarios that in the BAT scenario. This echoes the challenge of achieving the current capacity expansion plans of the EU[43] while simultaneously reducing the dependency on energy imports from outside the EU.

Like current natural gas imports, hydrogen imports will play a major role for the future European energy system and to reduce overall system costs. Repurposing existing natural gas infrastructure like pipelines or LNG terminals can potentially lower the associated cost of imports significantly[44]. North African countries are strategically positioned to competitively supply hydrogen to Europe as they have both high solar irradiation for renewable hydrogen production[45] and natural gas reserves for low-carbon hydrogen. In addition, they already have an existing natural gas export infrastructure to Europe. Although the amount of renewable energy needed for hydrogen exports from these countries to Europe is very limited compared to their land availability, ramping up the import capacities towards Europe can be challenging in practice[46]. Both hydrogen imports and distributed domestic production requires the development of a transnational hydrogen network for transporting hydrogen within Europe. This should be built around the repurposing of the existing natural gas network but will also require accelerated investments in purpose-built infrastructure[44]. Moreover, water consumption for renewable hydrogen production can limit the water availability in the exporting regions, that are already facing potable water scarcity issues[47,48]. However, hydrogen production via electrolysis in HyPE uses water produced from seawater desalination only in the areas within a reasonable distance from the seas (see Supplementary Information 2).

As a limit of this study, the cost for methane abatement was not endogenously included in the model, and hence, may lead to an unaccounted price increase for natural gas in BAT compared to CEF. However, the previously mentioned back-of-envelope calculation of the cost of methane abatement based on IEA's methane tracker cost data shows that it corresponds to 0.4% of the cumulative costs for natural gas in the 30 years' perspective in our modelling (and an even lower part of the overall energy system cost). Hence, not including the costs endogenously seems acceptable in combination with the general uncertainty in future natural gas prices. In addition, it was assumed that the trajectories for methane abatement were followed by the different countries simultaneously. A deviation by individual countries may lead to carbon leakage and give them an economic advantage that must be countered through the inclusion of upstream methane emissions in the European carbon border adjustment mechanism, as outlined previously.

Finally, while we analyse the role of the two main GHGs responsible of global warming as of today ($CO_2$ and methane), in a future with nearly a fourth of final energy consumption satisfied by hydrogen,

hydrogen leakage can also cause significant global warming effect (with 30 times higher potency than $CO_2$ over 20 years)[49]. In our analysis we use empirical data to assess the methane footprint of natural gas, and the technology data to assess $CO_2$ emissions. Such a data for hydrogen leakage can be available once there is a widespread hydrogen value chain. Nevertheless, as this hydrogen value chain is in its early development phase, best practices regarding the minimisation of hydrogen leakage can be put in place by the potential hydrogen suppliers, transporters and consumers to avoid repeating the historical mistake of the oil & gas industry.

## Methods

This study includes a coupled modelling framework representing the European energy system in a detailed manner with possibility of hydrogen imports from neighbouring regions. Moreover, methane footprint of natural gas and its derivative products are taken into account via the methane emission factor calculation module. The modelling represents the energy system from the primary energy source (crude oil, wind, solar, etc.) until the end-use energy form consumed (electricity, heat, etc.) in each sector (buildings, transport, industry and agriculture) including different primary energy supply options, energy transformation technologies, different primary and secondary energy carriers (electricity, hydrogen, coal, etc.) and energy storage options. The modelling framework is based on economic optimisation of the whole energy system including imports: minimisation of the cost of energy supply until consumption point over the outlook period (2016-2050). Therefore, taking into account the economic and technical characteristics of the overall energy system, the model finds the least-cost technology mix to meet the energy demand of each sector, considering sector coupling (interdependencies between different sectors, energy carriers and their supply options, and conversion technologies). Therefore, it takes into account both the competition over the scarce energy sources (such as Bioenergies) and the synergies of coupling between different sectors.

### The modelling framework

The modelling consists of the detailed European energy system optimisation model and a hydrogen delivery chain optimisation model that enables imports of hydrogen to Europe. The energy system optimisation model (MIRET-EU) represents the European energy system based on linear programming at country level, with a detailed representation of all technologies. The hydrogen delivery chain optimisation model represents the cost of hydrogen production and its export to Europe from neighbouring regions allowing competition between different clean hydrogen supply technologies (wind- or solar-based renewable hydrogen, low-carbon hydrogen based on methane reformation with carbon abatement and low-carbon hydrogen based on methane pyrolysis) and their origins: Europe, North African countries and Middle Eastern countries. Moreover, it includes the competition between different hydrogen transport routes (gaseous hydrogen via pipelines or liquified hydrogen and ammonia via shipping) to assess the least-cost hydrogen import routes. Via methane emissions calculation module, the modelling framework also accounts for the methane footprint of natural gas and low-carbon hydrogen, both produced in Europe and imported from neighbouring regions Europe.

### The MIRET-EU model

MIRET-EU is a dynamic energy system optimisation model that represents European energy system in a detailed manner from primary energy sources until the final energy in its consumed form. It optimises the investments in different energy supply, transformation, transport and storage through the period 2010 to 2050 to estimate least-cost capacity investment pathways based on the TIMES generator. It is a bottom-up economic model that combines technical engineering and economic approaches. The modelling considers linear programming to produce and results in a least-cost energy system across regions and sectors based on environmental (GHG emissions), political, resource-related and technical constraints over medium to long term. Such a modelling exercise gives important insights on long-term investment decisions in futuristic complex systems where technologies and energy infrastructure are set to evolve and vary from their current versions.

The modelling exercise is based on perfect market competition and perfect foresight regarding the future events, which means that the investment decisions are made based on full knowledge of future events with no information assymetry[50]. Findings of such a modelling exercise can support decision-making processes regarding the competivity of different technologies in the energy industry and can provide reliable grounds for the exploration possible futures of the European energy system following different policy, resource and technology scenarios. The modelling framework of the MIRET-EU model follows the evolution of the TIMES framework successively in the RES2020's Pan European TIMES model[51] and the JRC-EU-TIMES[52], MIRET-FR[53] and TIAM-IFPEN[18,54−57] models with additional technology-specific constraints in the transport sector, refineries and bioenergy conversion technologies, hydrogen infrastructure, power sector and industrial processes.

MIRET-EU studies the evolution of the European energy system in the period between 2010 and 2050 with 10-year optimisation time steps. As a part of this optimisation concerns past dates, the 2010–2020 period is calibrated and fine-tuned using the latest constraints (COVID-19 pandemic, Russia-Ukraine war, etc.) and data from the most well-known energy databases such as the JRC-IDEES (The EU's Joint Research Centre's Integrated Database of the European Energy System), POTEn-CIA (Policy Oriented Tool for Energy and Climate Change Impact Assessment), EUROSTAT, and other databases published by international organisation such as IEA, IRENA, and the World Bank.

For clarity in the final charts, we set the reference year to the previous mid-decade (i.e., 2016) for benchmarking comparisons with existing data in all sectors and all countries. The significance of the findings remains in the development of the European energy system in 2030 considering the current policies and latest political measures, but also beyond, in the year 2050. The model considers the existing technologies which are related to what is already installed in all considered countries in the historical period (i.e., the 2010s period), and the new technologies which are to be available in the future (e.g., from 2020 onwards). Spring, summer, fall, and winter are the four seasons considered by MIRET-EU, with each season broken down further into day, night, and peak resolution (it follows the same time-slice disaggregation as in the world multiregional model TIAM-IFPEN). Hence, each year is broken down into twelve time-slices that roughly correspond to the average daytime, night-time, and peak demand for each season (e.g., summer day, summer night and summer peak, etc.). The model is disaggregated into 27 countries (24 EU member states and 3 non-EU countries) Each country has its own energy system with its main demand sectors.

MIRET-EU provides in-depth, specific technological descriptions of each country's energy system from the primary resources (uranium, crude oil, coal, natural gas, biomass, renewables, and traded hydrogen) to the end-use sectors. It encompasses all stages through the chain of processes that transform, transport, distribute and convert energy into energy vectors and energy services. It includes the electricity generation technologies (all power plants from fossil-based to renewable energy sources) fuel production, primary and secondary energy sources, as well as imports and exports. Petroleum products, electricity, natural gas, hydrogen, and $CO_2$ captured may all be traded between countries. Energy is supplied to the demand side, which is composed of residential, commercial, agricultural, transportation, and industrial sectors, via various energy carriers (electricity, fossil fuels, e-fuels, heat, hydrogen, bioenergy, and other renewables). A package of measures and constraints such as quality standards for refinery products, the functioning

hours of power plants, global emission constraints or sectoral restrictions, has also been considered in MIRET-EU. More information on the MIRET-EU model and the considered sectors and technologies as well as the cost parameters can be found in Supplementary Information 1.

## The HyPE model

HyPE is a hydrogen delivery chain optimisation model with high spatial (0.5°) and temporal resolution (hourly time slices for at least one full year) grounded on the literature on international hydrogen trade[45,58–60]. The model finds cost-efficient pathways to balance point-to-point hydrogen demand. By representing the competition between hydrogen production technologies, production locations, the costs incurred for transport and logistics, and other abatement costs (i.e., $CO_2$ and methane emission), the model finds optimal investments in production technologies, use of resources, domestic supply and hydrogen tradeflows[18].

Following the European hydrogen strategy, only low-carbon and renewable hydrogen imports are considered, with a focus on the closest neighbouring regions: North Africa, the Middle East and Ukraine. The calculation of renewable hydrogen is as follows: The neighbouring regions were divided in a 0.5° grid where solar, wind, biomass and natural gas resources are assessed to compute levelized costs of hydrogen (LCOH—the levelized cost of hydrogen adopts the life cycle costing methodology where all related costs and produced quantities are included to compute an average ratio of cost per kilogram produced):

$$LCOH_{tech,y,country} = \frac{CAPEX_{tech,y} + \sum_{t=1}^{lt_{tech}} \frac{OPEX_{tech,y}}{(1+WACC_{tech,y,country})^t}}{\sum_{t=1}^{lt_{tech}} \frac{E_{tech,cell}}{(1+WACC_{tech,y,country})^t}} \quad (1)$$

$$E_{tech,cell} = \sum_{h=1}^{8760} CF_{h,tech,cell} \times \frac{1}{\eta_{electrolysis}} \quad (2)$$

Where:

| | |
|---|---|
| $CAPEX_{tech,y}$ : | Initial investments for a given production technology *tech* on year *y* |
| $OPEX_{tech,y}$ : | Maintenance and operational costs for a given *tech* and year *y* |
| $WACC_{tech,y,country}$ : | Weighted Average Cost of Capital in the *country* and year *y* per *tech* |
| $E_{tech,cell}$ : | Annual energy output per *tech* on a production *cell* in kilograms of hydrogen |
| $CF_{h,tech,cell}$ : | *Capacity Factor* — Energy produced out of one kW of capacity installed, in kWh, per hour *h*, *tech* on a production *cell* |
| $\eta_{electrolysis}$ : | Consumption of electricity of the electrolyser in kg/kWh |
| $lt_{tech}$ : | Lifetime of the production technology *tech* considered |

Adding the transport options between different countries (considering the retrofitted natural gas pipelines and maritime transport via liquified hydrogen and ammonia including the needed conversion and reconversion costs) the transport routes are identified and optimised. The hydrogen import potential in Europe is obtained in the form of a supply curve for each possible delivery point.

Following an economic optimisation, the competition between different hydrogen export options to Europe including potential big demand hubs (such as China and Southeast Asia) has been assessed to mimic the potential competition on hydrogen resources. The import supply curves are used by the energy system model to represent the competition between European domestic production and imports. More information on HyPE and calculation of LCOH for green and blue hydrogen can be found in Supplementary Information 2.

## Methane emission scenarios and emission factor calculation

Methane emissions along the natural gas value chain were estimated for over 30 countries, including upstream, downstream and LNG processes where relevant. The methane emission scenarios were developed to provide scenarios extending from baseline to best case scenario. The first scenario, Current Emission Factors (CEF), represents

the current best understanding level of emissions in the countries. Information from country level academic papers[61,62], with significant measurements and estimates, emissions reported to UNFCCC[63] and IEA methane tracker[64] were leveraged to develop a concrete understanding of emissions from the gas sector in each country. A decision matrix was used to select the best source of information in each case (please refer to Supplementary Information 3 for further information). Following this, the emissions from the different steps of the natural gas value chain were summed up separately and divided by the volume of natural gas produced, transported, or imported, to estimate the EF of each step. This scenario, called Current Emission Factors (CEF), presents the average EF per country and forms the "baseline" for other scenarios to be developed from. The EF is considered flat through the period until 2050 in this scenario.

The second scenario, Harmonised Pledges (HP), is the intermediary scenario between current emission factors and an optimistic best available technology deployment scenario. It includes the effect of country policies[34,35], NDCs[36] relevant to GHG emission reductions and international pledges[37–40] (International Pledges such as: (a) Zero flaring by 2030[37] (b) Methane Pledge[38] (c) Methane Alliance[39]) published as of 2023. In this scenario, the changes in the EFs from the preliminary year (2019) to 2050 are assessed based on the announced policies and pledges. Depending on their policies and participation in international pledges, countries are grouped in four categories: countries with relevant GHG emission reduction policy and international pledges, countries with relevant GHG emission reduction policy but without international pledges, countries without policy relevant for GHG emission reduction and only international pledges and countries with no policy nor international pledges. For the first category, we apply the policy reduction goal by the target year and then, the optimistic target within international pledges. For the countries in the second category, the existing policy or NDC target is applied by the target year and the EF remains flat beyond the policy date until 2050. We apply the Zero Routine Flaring to the countries in the third category by 2030 and the less optimistic target among the signed international pledges beyond 2030. The EFs of the countries in the last category, however, remain flat between 2019 and 2050.

The final scenario, Best Available Technologies (BAT), represents the EFs that could be achieved assuming BAT deployment along the natural gas value chain by a certain year. The number of years required to reach this scenario's EFs changes by country, however a minimum of 10 years is assumed to reach BAT EF for all countries. IEA abatement potential[64], industry targets set in the countries assessed, or global industry targets[65] were considered to estimate the impacts on EF from the preliminary year (2019) to 2050. Either IEA's methane tracker data or OGCI industrial targets were used to identify the EFs for this scenario, depending on the data availability. A more detailed explanation of methane emissions calculation methodology and the scenarios can be found in Supplementary Information 3.

The calculated methane emission factors are implemented in MIRET-EU for the three scenarios (BAT, HP and CEF). Under the GHG emission reduction constraints, the evolution of the energy system is optimised independently for each of the three scenarios, assuming that the measures deployed to limit methane emissions are taken for granted. Hence, the investments needed to reach lower methane EF along the natural gas value chain are exogenous to the optimisation but are evaluated *ex post* for each scenario and compared to the achieved cost reduction of the whole energy system when lower EF are implemented in the optimisation (see the Discussion section).

## The choice of global warming potential period for methane leakage

The global warming potential of methane should be assessed considering the horizon of the climate target[66]. For a climate-neutrality objective in 2050, the methane concentration in the atmosphere

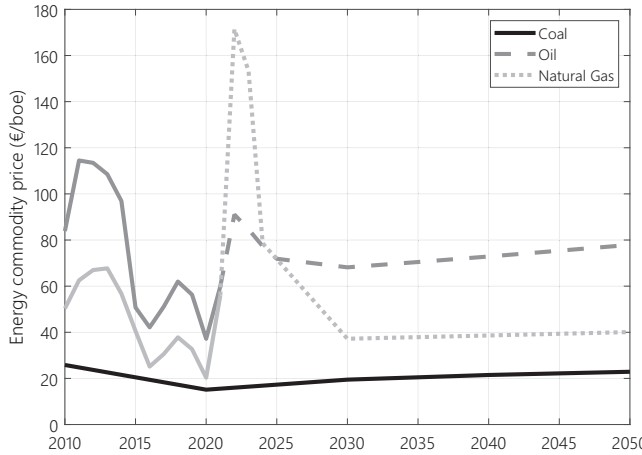

**Fig. 7 | Fossil fuel prices trajectories implemented in the modelling.** Short term oil and natural gas prices are expected to peak in reaction to the war in Ukraine and of the supply restriction. Long term prices paths follow the IEA STEPS scenario[69]. STEPS is one of three key scenarios of the International Energy Agency, standing for stated policies.

should be stabilised by 2045, and the period from the assessment date (2023) to this date (2045) should be considered as reference global warming potential assessment period, which is 22 years. Therefore, we chose $GWP_{20}$ as the reference for translation of methane emissions in $CO_{2eq}$ terms.

## Carbon capture and storage potential

The annual $CO_2$ storage capacity is assessed based on the values suggested in the existing literature. The assessment considers two key parameters: total available geologic potential (overall European cumulative $CO_2$ storage capacity) and the dynamic rate of increase in annual $CO_2$ storage capacity (based on drilling and sequestration capacities). The United Kingdom[18] and Norway[18] contain the largest $CO_2$ storage potentials in Europe each with offshore storage capacity as high as 70 $GtCO_2$. Regarding the annual injection rates, the analysis of potential for scaling up $CO_2$ storage on the Norwegian continental shelf shows that only Norwegian offshore wells have an average injection rate capacity of $0.695 \pm 0.222$ Mt/year[67]. Following the analysis carried in Hydrogen for Europe study[68], we estimate that only in Norway, 2083 wells could be active by 2050 which corresponds to an available injection rate of about 1.4 $GtCO_2$/year by 2050. Following the technical, political and social acceptability uncertainties on the development of carbon capture and storage, the maximal annual $CO_2$ storage capacities are set to 1 $GtCO_2$/year in 2040 and 1.4 $GtCO_2$/year by 2050 across Europe.

## Policy assumptions

Several assumptions made in the modelling are key to understand the observed evolution of the European energy system. The most structuring ones concern the energy policies adopted and proposed at the European level, which are assumed to be mandatory in the model. For the $CO_2$ and methane emissions from natural gas value chain, an overall progressive reduction is implemented. Following the European Fit-for-55 directive, a 55% reduction of GHG emissions by 2030 (compared to 1990 levels) and reaching net-zero by 2050 targets are implemented. Sectoral GHG emission reduction goals are also included for both EU ETS (reduction by 21% in 2020 compared to the 2005 levels including aviation and by 43% in 2030) and non-ETS (ranging between 0% and −40% in 2030 compared with 2005 levels in order to achieve a collective 30% reduction of the total EU emissions) sectors, as well as for passenger transport (target for the EU fleet-wide average emissions is set at 95 g$CO_2$/km).

Regarding the share of renewable energy in gross final energy consumption, a target of 40% is set for 2030, complemented by a specific target for the transport sector, following the revised RED II directive of the European Commission: for the period 2030 to 2050 the target is set to 14%. The contribution of biofuels produced from food and feed crops (1st Generation) is capped at 7% of road and rail transport fuel in each Member State from 2020 onwards. Furthermore, the contribution of advanced biofuels and biogas (2nd Generation) should be at least 0.2% in 2022, at least 1 % in 2025 and at least 3,5 % in 2030 (as a share of the final consumption of energy in the transport sector). Finally, an objective of 32.5% energy efficiency in 2030 compared to a European business-as-usual scenario is also considered. All these targets for emission reduction, renewable energy share and energy efficiency are aligned with European climate goals: Fit-for-55, European Green Deal and European climate legislation.

More recently, the REPowerEU plan announced a strategy to phase out energy supply from Russia for geopolitical and sovereignty reasons. To reflect this strategic independence decision, a zero Russian import constraint from 2025 is implemented for natural gas and hydrogen in the modelling. For coal and oil, no specific supply restriction is included as their contribution to Europe's energy supply is anyway reduced following the emission reduction constrains.

## Data assumptions

Naturally, a direct impact of the Russia-Ukraine conflict and associated supply restrictions on fossil fuel prices can be expected. Therefore, the initial prices trajectories based on the STEPS (Stated Policies) scenario of the IEA World Energy Outlook 2021[68] are adjusted in the short-term: oil ICE and natural gas TTF future prices are used to represent the expected peak in prices of these commodities between 2022 and 2025, before stabilising on the STEPS price path by 2030, that will follow until the end of the modelling window (see Fig. 7). Coal prices remain unchanged as this commodity is the fastest to be phased out and its prices are less dependent on imports.

The hydrogen production technologies and the related cost and efficiency data can be found in the annex E (Data documentation, p.162-164) in the Hydrogen for Europe study[68]. More detailed information on taken assumptions and the overall database can be found in Supplementary Information 1.

## Reporting summary

Further information on research design is available in the Nature Portfolio Reporting Summary linked to this article.

## Data availability

The modelling output data generated in this study have been deposited in the H4EU-Methane repository in the following link: https://doi.org/10.5281/zenodo.8149950.

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

## Acknowledgements

This study was funded by International Association of Oil and Gas Producers (IOGP), as a part of the Hydrogen4EU project. The authors are very grateful to François-Régis Mouton and Axel Scheuer for the funding and support and to Malcolm Rice-Jones, Rosanna Fusco, Hannah Davies, Frank Winkelhoff, Julia Schmitt and Nareg Terzian for their very useful comments and insights. The authors would also like to thank Johannes Brauer, Clément Cabot, Lisa Ronce and David Ah-Voun for their contribution to the study.

## Author contributions

J.T., S.D., J.Sa., E.H. and S.S. developed the research idea and G.S.S., M.Vi. and B.S. developed the modelling framework. G.S.S., M.Vi. and C.B. conducted the simulation and B.S., S.D., E.H., G.S.S., J.Sa., V.D.H., L.M.M., G.A.R. and J.St. analysed the findings. M.Ve., F.L. and S.S. developed the methane emissions calculation framework, and M.Ve. and F.L. calculated the methane emissions factors. J.T. and S.D. were responsible of general project management, while B.S. was the responsible of the paper drafting and its organisation. Moreover, J.T. secured the project funding. L.M.M. produced the figures of the manuscript and B.S., E.H., L.M.M, V.D.H., G.S.S., M.Ve. and J.St. wrote the manuscript, that was completed by the whole consortium's revision.

## Competing interests

The authors declare no competing interests.
