## [Peer Review File · Nature Communications]

REVIEWER COMMENTS

Reviewer #1 (Remarks to the Author):

This paper addresses a timely and relevant question. How could the European Union reduce its carbon dioxide and methane emissions to achieve net-zero greenhouse gas emissions by 2050 at the lowest possible costs? To investigate this question, the authors use an energy system model that minimizes costs over multiple periods until 2050, with constraints on the equivalent carbon dioxide emissions (including both CO₂ and CH₄) and over a set of scenarios. The results show that fossil gas demand could significantly differ depending on methane emissions along the supply chain.

Although the paper is largely correctly written, the logical structure is deficient. Some elements (e.g., the different "pathways") appear out of the blue in the middle of the paper. Moreover, major decisions that influence the analysis results, such as the exclusion of Russian energy exports and quotas for the contribution of renewables, are insufficiently motivated and explained. The method description in the main text is too shallow, and the one provided in the supplementary material could be significantly improved. In particular, more transparency about the model results is necessary.

I cannot recommend the publication of this paper in its current form.

In short, the paper tries to do too much at once. This results in a confusing paper, with shallow explanations of the method and results and a very unsatisfactory discussion of the implications. In my view, the number and depth of changes needed to recommend the publication of this paper in Nature Communications exceed what is reasonable to ask from the authors in a review process. Therefore, my recommendation to the Editor is to reject the paper, and to the authors, I recommend focusing on a single aspect of their study and fundamentally rewriting the piece. I believe the core contribution is found in the sections until line 154 and the implications outlined in lines 233-243, and I suggest focusing on them for rewriting the paper.

These are my general criticism of the paper:

1. Insufficient detail to assess if the analysis supports the paper's conclusions. The results are somewhat surprising because efforts to limit methane emissions in the supply chain led to more fossil gas used in the energy system. Therefore, this suggests fossil gas remains a cost-effective energy option even when the best-available techniques are used to reduce methane emissions.

This raises two questions that are fundamental to the core contribution of the paper. First, what is the price of fossil gas in the model, and how does it change until 2050? In the supplementary information, the authors mention an assumption of \$2.4/MMBTU production costs at the wellhead and further explain their approach in footnote "qq". This cost of fossil gas production is lower than historical gas prices, between \$3-8/MMBTU based on the Dutch TTF natural gas future prices, and much lower than current gas prices in Europe of around and above \$30/MMBTU. Transportation and insurance expenses might take the price of fossil gas in the model to the lower end of historical prices in Europe. However, given the assumption that Russian exports will not be resumed, how do the authors justify gas prices on the lower end of the historical range?

Second, how do the measures to reduce methane emissions in the supply chain affect fossil gas prices? I could not find a discussion of whether the costs of reducing methane emissions are considered. One would expect that more stringent measures to reduce methane emissions would make fossil gas costlier and thus move the results in the BAT scenario towards less fossil gas use. The description of the method to calculate emission factors in the supplementary information only states that measures that do not incur additional costs are assumed to be adopted earlier than those that impose additional costs on firms in the fossil gas supply chain. The cost of the measures is referred to in the discussion's second paragraph, but it remains unclear whether those costs are factored into the price of fossil gas in the optimization model.

An important question that authors should address in the discussion is carbon leakage. Although all countries are assumed to follow each scenario specification, in the real world, some countries may impose stricter controls on methane emissions than others and do so at different times. For example, suppose more stringent controls lead to higher fossil gas prices. In that case, a cost-minimizer decision-maker may increase imports to Europe from producers with "dirtier" methane supply chains and reduce those from producers with "cleaner" supply chains. What are the implications? Are there mechanisms in place to prevent this? Should there be? Which ones? What can the authors discuss based on their results?

2. Major assumptions that are insufficiently motivated and explained. There are several major decisions that the authors do not motivate or explain: (1) the exclusion of Russian energy imports, (2) the use of quotas for renewable energy, (3) the use of 2016 as the reference year, and (4) many detailed hydrogen model assumptions.

There are too many detailed assumptions concerning the hydrogen model that the authors would need to justify better to list them here. However, if the manuscript goes through a second revision round, I will address them in detail. They include questions mainly about input data and constraints of the model.

In addition, recent literature on hydrogen trade contradicts some of this paper's findings. The difference in results is likely to be driven by some of the assumptions in the hydrogen model. The authors do not discuss this.

3. Inadequate discussion of implications. Outside of the second paragraph, the discussion section reformulates analysis results (or introduces new results, as with figure 8) without discussing the implications of the results in depth. Instead, the authors should deal with the implications of these results.

An additional minor comment about presentation. The use of references is not consistent; there are duplicated references in the supplementary information and formatting and grammar errors. All of these must be adequately addressed.

Overall, there is insufficient evidence to assess the soundness of the results, there are questionable decisions in the method (in particular about the hydrogen model), and there are important flaws in the presentation and discussion of the results in the main text. Despite addressing a relevant question and presenting a novel analysis, I cannot recommend the publication of this paper as it is.

Reviewer #2 (Remarks to the Author):

Most studies of the transformation of energy systems concentrate on carbon dioxide abatement. This paper extends a model of the European energy system to also include methane abatement jointly with CO₂. Including methane leads to higher uptake of renewable energy. The role of natural gas in the energy system depends on abatement measures taken up and down the natural gas value chain.

The paper has interesting and novel results. It is well written, both in terms of language and arguments. I have some comments and concerns about the modelling methodology which I would appreciate the authors to address.

- My understanding of the Integrated Assessment Models used e.g. in IPCC summary reports, is that they do include all GHG, not just CO₂. I'm not sure to what extent interlinkages with the energy system are endogenously modelled. Perhaps the authors could clarify this point in the introduction.

- The choice of using GWP20 and GWP100 is not that well justified, but has a big impact on the study results. The authors write "In the context of this assessment, aiming for GHG-neutrality by 2050, GWP20 has been given priority." But climate change is also a long-term problem, with most damages occurring after 2050. Choosing GWP20 over GWP100 leads to a much larger impact of methane on the study results, which is beneficial to the argument of the paper of course. Perhaps the authors could do a single sensitivity analysis with GWP100?

- Hydrogen also has a GWP20 of ~30 and GWP100 ~8 (see e.g. <https://doi.org/10.1016/j.ijhydene.2022.11.219>). Has this also been included, given that the paper results see nearly a quarter of final energy from hydrogen in 2050?

- What is the reason for so much blue hydrogen in 2030? Is it cost driven or are their limits in the model for the scale-up rates of electrolysis?

- I didn't understand how the capture rates are treated for blue hydrogen. The SI mentions a rate of 95%, which is much higher than current facilities achieve. Then the text says "were accounted for by assuming economic offsets". Why does unabated CO2 not go into the model's CO2 balance?

- Is net-zero emissions for CO2 and CH4 appropriate? Doesn't it depend on what is going on in agriculture and LULUCF sectors? It could be net-positive or net-negative depending (assuming an overall net-zero GHG target).

- Are the costs of applying BAT technologies included in the total system costs? In the discussion it says "In contrast, the costs for achieving BAT deployment in the natural gas value chain are below 1% of the additional cost of the CEF scenario", so I wasn't sure.

- Do downstream costs include leakage in distribution grids or end-use devices?

- From Figure 4 I was unclear what is going on with aviation and maritime fuels. The "feedstock for transport" seemed too small to cover international travel with e-fuels.

- Figure 8: 1.4 GtCO₂/a sequestration is pretty high, compared e.g. to JRC-EU-TIMES scenarios and others. For a technology with a couple of MtCO₂/a at the moment. I'm not suggesting the authors change anything, it's just a comment.

- Methods section: please mention MIRET-EU only has 12 time slices per year. For HyPE, please be more concrete with "high spatial and temporal resolution" since this is relative.

- SI Figure 8: Very nice that you included country-specific WACC! You see it in difference of LCOH in Saudi Arabia versus Yemen.

- For solar 170 MWp/km² is a density for the panels themselves. PV farms rarely reach above 70 MWp/km².

Response to referees

A game changer? The impact of methane leakage on the role of natural gas in the European energy transition

Reviewer #1

First, we wish to thank warmly the reviewer for the insightful comments raised in her/his report that led to the revision of our paper based on her/his recommendations. Please note that we completely agree with the need for a fundamental restructuring of the paper, and we took the reviewer's advice. While we highlight the changes below, the structure of the paper has been fundamentally modified, notably by adding the gross final energy consumption to "Highly renewable future, with different futures for natural gas", modifying the sections "Each pathway needs a different emission offset" and "Hydrogen: one of the key pillars of reaching climate neutrality" marginally and modifying the remaining fundamentally. We have pasted the comment in the first column and indicated how we have taken the comments into account in the second one. New text inserted in the revised version of our manuscript is indicated in *bold italic*. The manuscript and the supplementary information files are provided in two files: track-changes (with modification in red) to track these changes and clean version.

Reviewer comment	Answer
Some elements (e.g., the different "pathways") appear out of the blue in the middle of the paper.	Many thanks for highlighting this lack of our manuscript. We tried to detail each of the new elements and provide the background for them. Following your insight, we dropped the sensitivity on the renewable share that we had called pathways, and we don't discuss any renewable targets (Renewable Push pathway). following restructuring the article, we believe that it's all clearer now.
Moreover, major decisions that influence the analysis results, such as the exclusion of Russian energy exports and quotas for the contribution of renewables, are insufficiently motivated and explained.	We agree with the limited justification level of our strong assumptions. We addressed them one by one in your more detailed comment regarding the subject below.
The method description in the main text is too shallow, and the one provided in the supplementary material could be significantly improved. In particular, more transparency about the model results is necessary.	Please find a detailed description both in methods section and in the Supplementary Information files. To be more precise, we separated the supplementary information files, and each file is now adapted to your comments, with its separate references and more technical description. Regarding the body of the main paper: 1. The description of MIRET-EU model is detailed further in the methods section by more than doubling the text volume in the Methods section in pages 9 and 10.2. The description of the HyPE model in the methods section is detailed further by adding its temporal and spatial granularity, included countries and the main technologies and the baseline of

	the LCOH calculation (Equation 1.a and Equation 1.b), in pages 10 and 11.  3. We also added further information in the methods section by adding the taken baseline assumptions, carbon capture and storage potential and the choice of GWP20 for the conversion of methane emissions to CO_{2eq} terms, in pages 13, 14 and 15. 4. We also added a Data Availability section, where we refer to the first author's GitHub page for the data used in the production of the graphs – modelling results.
I believe the core contribution is found in the sections until line 154 and the implications outlined in lines 233-243, and I suggest focusing on them for rewriting the paper.	We agree, and we dropped the section “Going more renewables” in the results presentation. We also dropped the discussions related to the results of this section. Moreover, we reduced the hydrogen and trade results and the related discussion, and we increased the discussion on the energy mix, the gross final energy consumption, methane emissions and their impact on the role of natural gas. Discussion section is completely reformulated (pages 7, 8 and 9).
Insufficient detail to assess if the analysis supports the paper's conclusions. The results are somewhat surprising because efforts to limit methane emissions in the supply chain led to more fossil gas used in the energy system. Therefore, this suggests fossil gas remains a cost-effective energy option even when the best-available techniques are used to reduce methane emissions. This raises two questions that are fundamental to the core contribution of the paper:  - First, what is the price of fossil gas in the model, and how does it change until 2050? In the supplementary information, the authors mention an assumption of \$2.4/MMBTU production costs at the wellhead and further explain their approach in footnote "qq". This cost of fossil gas production is lower than historical gas prices, between \$3-8/MMBTU based on the Dutch TTF natural gas future prices, and much lower than current gas prices in Europe of around and above \$30/MMBTU. Transportation and insurance expenses might take the price of fossil gas in the model to the lower end of historical prices in Europe. However, given the assumption that Russian exports will not be resumed, how do the authors justify gas prices on the lower end of the historical range? 	In the model, we implement the 55% reduction target and neutrality to CO₂ and CH₄ for 2030, and a full neutrality target for 2050. The constraint of those emissions is driving the share of fossil fuels. The cost of methane abatement is lower than its social cost, and also smaller than the overall cost increase of the system to respect these constraints without abatement (based on IEA's methane tracked database). Therefore, considering we have the same constraint applied to all scenarios, when we have high methane abatement, the model has the possibility to use slightly more fossil fuel with existing technologies or new fossil-based technologies which are cheaper (optimisation of the model). The answers to the highlighted questions are as below:  - We consider the natural gas prices aligned with IEA's stated Policies Scenario as of the beginning of 2022 (at the moment of the analysis). The natural gas prices are going back to the pre-crisis levels¹. That's also what we have considered in the natural gas prices' assumptions: based on the TTF futures, we considered that the natural gas prices will be high and peak before 2025 and

¹ <https://tradingeconomics.com/commodity/eu-natural-gas>

-Second, how do the measures to reduce methane emissions in the supply chain affect fossil gas prices? I could not find a discussion of whether the costs of reducing methane emissions are considered. One would expect that more stringent measures to reduce methane emissions would make fossil gas costlier and thus move the results in the BAT scenario towards less fossil gas use. The description of the method to calculate emission factors in the supplementary information only states that measures that do not incur additional costs are assumed to be adopted earlier than those that impose additional costs on firms in the fossil gas supply chain. The cost of the measures is referred to in the discussion's second paragraph, but it remains unclear whether those costs are factored into the price of fossil gas in the optimization model.

come back to the projected values in IEA's Stated Policies scenario. The wellhead natural gas price is used for the calculation of the blue hydrogen production cost in the exporting countries. For the European natural gas case, the values are projected between €/MMBtu and €/MMBtu (topping at around €30/MWh), which are close to historical natural gas prices, and in-line with current fall of the price of natural gas in Europe (€13/MWh). We tried to make it more clear in the methods section, by adding the figures showing the projections of the natural gas, crude oil and coal prices and the corresponding sources in page 15, Figure 7.

- Indeed, more stringent measures to reduce methane emissions would make fossil gas costlier and less fossil gas use. Nevertheless, in our case, we applied only one common constraint on CH₄ emissions "budget", but with three different evolutions of emission factors. It means that in the model, we implement the 55% reduction target and neutrality to CO₂ and CH₄ (by 2030). The constraint of those emissions is what drives the share of fossil fuels. Considering we have the same constraint applied to all scenarios, when we have a methane abatement, the model has the possibility to use slightly more fossil fuel with existing technologies or new fossil-based technologies which are cheaper. We don't optimise the methane abatement adoption, but rather we define three scenarios of emission factors, and we optimise the model based on them.

In the updated Discussion and conclusion section, we tried to make it clearer for the readers, notably via addition of the following text in the end of page 8 and beginning of page 9:

The cost for methane abatement was not endogenously included in the model, and hence, may lead to an unaccounted price increase for natural gas in BAT compared to CEF. However, a back-of-envelope calculation

	of the cost of methane abatement based on IEA's methane tracker cost data shows that the cost for methane abatement is only a small fraction of the cost of natural gas. In fact, it corresponds to 0.4% of the cumulative costs for natural gas in the 30 years' perspective (below 1% of the overall energy system cost). Hence, not including the costs endogenously seems acceptable in combination with the general uncertainty in future natural gas prices.
An important question that authors should address in the discussion is carbon leakage. Although all countries are assumed to follow each scenario specification, in the real world, some countries may impose stricter controls on methane emissions than others and do so at different times. For example, suppose more stringent controls lead to higher fossil gas prices. In that case, a cost-minimizer decision-maker may increase imports to Europe from producers with "dirtier" methane supply chains and reduce those from producers with "cleaner" supply chains. What are the implications? Are there mechanisms in place to prevent this? Should there be? Which ones? What can the authors discuss based on their results?	We agree that the topic of carbon leakage is an important part of discussing the implications of methane abatement, and we consider a synchronised behaviour among the considered countries. To discuss this heterogeneity issue, we modified the discussion and added a new paragraph related to carbon leakage and the potential for avoiding it. To this end, we suggest including it as well in European carbon border adjustment mechanism. The following text was added to the Discussion and conclusion section, in page 9, paragraph 1, lines 3 to 7: In addition, it was assumed that the trajectories for methane abatement were followed by the different countries simultaneously. A deviation by individual countries may lead to carbon leakage and give them an economic advantage that must be countered through the inclusion of upstream methane emissions in the European carbon border adjustment mechanism.
Major assumptions that are insufficiently motivated and explained. There are several major decisions that the authors do not motivate or explain: (1) the exclusion of Russian energy imports, (2) the use of quotas for renewable energy, (3) the use of 2016 as the reference year, and (4) many detailed hydrogen model assumptions.	We agree that our strong assumptions are not clearly justified in the text. We take all of your points:  1. Exclusion of Russian gas is one of the central points of this study, where we analyse the implications of not having pipeline imports of neither natural gas nor hydrogen from Russia. With no surprise, this comes at the cost of increased LNG imports, and therefore, increased natural gas price. Moreover, it also reduces the natural gas consumption of the Europe from a baseline with more cheap natural gas available, and it increases the pace of the energy transition. The cheap low-carbon hydrogen availability is also lessened and Europe imports renewable hydrogen from other neighbouring regions, mainly from North Africa and Middle East and this increase also comes with its extra cost. The reason why we exclude Russian gas is the European policy objectives following Russian invasion of Ukraine:

REPowerEU plan, which set zero natural gas imports from Russia by 2030, and it also fixes a 16 Mt of clean hydrogen and 4 MtH_{2eq} of clean ammonia demand by this date, of which half should be produced locally in Europe, remaining satisfied via imports. This is also fully aligned with our modelling results (without any constraint on these values). To clarify the assumption made, we added the following explanation in the Methods section, where we explain baseline assumptions, in page 15, second paragraph, lines 1 to 4 as below:

More recently, the REPowerEU² plan announced a strategy to phase out energy supply from Russia for geopolitical and sovereignty reasons. To reflect this strategic independence decision, a zero Russian import constraint from 2025 is implemented for natural gas and hydrogen in the modelling.

2. The use of quotas for renewables is followed by the European policy ambitions: The revised RED II directive that fixes a 40% of renewable share in final energy consumption, and the further revision proposal by the Commission (as part of REPowerEU plan in May 2022) to increase the target to 45% of renewables in the final energy consumption³. As we dropped the scenario with higher renewable quotas (Renewable Push pathway), our explanation in the text focuses on the baseline quota considered in the RED II directive. To clarify it, we added the following clarification to the manuscript in the Methods section, page 15, paragraph 1, lines 1 to 4:

Regarding the share of renewable energy in gross final energy consumption, a target of 40% is set for 2030, complemented by a specific target for the transport sector, following the revised RED II directive of the European Commission.

3. The model is designed to investigate the evolution of the energy system from 2010 to 2050 with 10-year time-steps. For clarity in the final charts, we set the reference year to the previous mid-decade (i.e., 2016) for benchmarking

² https://commission.europa.eu/strategy-and-policy/priorities-2019-2024/european-green-deal/repowereu-affordable-secure-and-sustainable-energy-europe_en

³ [https://www.europarl.europa.eu/thinktank/fr/document/EPRS_BRI\(2021\)698781](https://www.europarl.europa.eu/thinktank/fr/document/EPRS_BRI(2021)698781)

	comparisons with existing data in all sectors and all countries. The significance of the findings rests in the development of the European energy system in 2030 considering the current policies and latest political measures, but also beyond, in the year 2050. The model considers the existing technologies which are related to what is already installed in all considered countries in the historical period (i.e., the 2010s period), and the new technologies which are to be available in the future (e.g., from 2020 onwards). The data for all of the studied sectors and subsectors and in all European countries are not available for 2022 yet, and the difference between 2016 and 2019 is very marginal that is not worth the whole data gathering exercise for a very small (invisible up to 1 percentage point) period. Moreover, the disaggregated data was not available for the year 2019 at the time of the analysis (by early 2022). The 2010-2022 period is calibrated using up-to-date constraints (COVID-19 pandemic, Russia-Ukraine war, etc.) and the most recent data from energy statistics databases such as the JRC-IDEES, POTEnCIA, EUROSTAT, and other international databases from IEA, IRENA, and World Bank, among others. We added this reasoning in the MIRET-EU model description in Methods section in page 11, first paragraph, and in Supplementary Information 1. 4. We tried to detail further the HyPE model assumptions in the new Supplementary Information 2. As the modification is fundamental, we can't use this space here to highlight all of them, nevertheless, the changes are highlighted in the revision file with track changes. We hope that it sheds enough light for its description.
There are too many detailed assumptions concerning the hydrogen model that the authors would need to justify better to list them here. However, if the manuscript goes through a second revision round, I will address them in detail. They include questions mainly about input data and constraints of the model. In addition, recent literature on hydrogen trade contradicts some of this paper's findings. The difference in results is likely to be driven by some	We fundamentally modified the description of the HyPE model and its inputs in Supplementary Information 2, and we detailed further the HyPE model description in the Methods section. Our modifications include further details regarding the cost data, technology constraints and the main equations.

of the assumptions in the hydrogen model. The authors do not discuss this.	
Inadequate discussion of implications. Outside of the second paragraph, the discussion section reformulates analysis results (or introduces new results, as with figure 8) without discussing the implications of the results in depth. Instead, the authors should deal with the implications of these results.	We agree that we have not discussed all implications of the results. To this end, we entirely changed the structure of the discussion section with an increased focus on the implications of methane emission reduction on the modelling outcome. We also dropped the new graphs that introduced further new information on the modelling results (the carbon capture and its source). The discussion includes hence now the impact of methane abatement on the overall natural gas cost, supply and risk, the potential for carbon leakage as well as the implications towards importing significant amounts of hydrogen and the need for anticipation of potential hydrogen leakage and using best practices to minimise it, in pages 8 and 9.
An additional minor comment about presentation. The use of references is not consistent; there are duplicated references in the supplementary information and formatting and grammar errors. All of these must be adequately addressed.	We reviewed the Supplementary Information file, we divided it in 3 files, one on the energy system model (MIRET-EU), one on the hydrogen imports model (HyPE) and one on the methane emissions calculation. We double-checked the grammar; we modified the references and we made sure that they are all in APA format.

Reviewer #2

First, we wish to thank warmly the reviewer for the insightful comments raised in her/his report that led to the revision of our paper based on her/his recommendations. Following the fundamental changes needed by the first reviewer's comments, we modified the paper deeply. While the first half of it remains nearly untouched, from the section on the presentation of the results on hydrogen supply and demand onwards the paper is modified fundamentally to focus more on the impact of methane leakage and its abatement on the role of natural gas. We dropped the scenario with high renewable quotas (Renewable Push pathway), and we separated the Supplementary Information files with higher details on the description of hydrogen imports model (HyPE). Following both reviewers' comments, we also modified the methods section to detail further the models and precise some of the strong assumptions. We have pasted the reviewer's comments in the first column and indicated how we have taken the comments into account in the second one. New text inserted in the revised version of our manuscript is indicated in ***bold italic***. The manuscript and the supplementary information files are provided in two files: track-changes (with modification in red) to track these changes and clean version.

Reviewer comment	Answer
My understanding of the Integrated Assessment Models used e.g. in IPCC summary reports, is that they do include all GHG, not just CO2. I'm not sure to what extent interlinkages with the energy system are endogenously modelled. Perhaps the authors could clarify this point in the introduction.	The definition and the differences between IAMs and energy system models can sometimes be very unclear, and they might seem very limited in description. The IAMs are global integrated models that use linkage of several sectors, notably the environment – energy – economy nexus. Nevertheless, these models are not granular enough regarding the technical precision that energy system models require and the micro-level questions such as chemicals industry's methanol subsector, or transport sector's aviation subsector, as well as the temporal and spatial constraints (such as variability of renewables, hourly variability of demand, the country-specific specificities) cannot be assessed with those models. Moreover, IAMs are mainly top-down General or Partial Equilibrium Models, and they do not include bottom-up optimisation (except few that are coupled with bottom-up models, such as MESSAGE) of the whole energy system. While IAMs consider the most important GHGs, the energy system optimisation models generally do not go to such a precision in upstream processes. To make it clear, we added following addition to the introduction section in the introduction section, in page 2, paragraph 3 and lines 3 to 7: While integrated assessment models (IAMs) generally include different GHGs, they lack the detailed representation of the energy system and temporal and spatial precision, and they are generally based on top-down allocation of energy sources and carriers and do not result from explicit optimisation.

The choice of using GWP20 and GWP100 is not that well justified, but has a big impact on the study results. The authors write "In the context of this assessment, aiming for GHG-neutrality by 2050, GWP20 has been given priority." But climate change is also a long-term problem, with most damages occurring after 2050. Choosing GWP20 over GWP100 leads to a much larger impact of methane on the study results, which is beneficial to the argument of the paper of course. Perhaps the authors could do a single sensitivity analysis with GWP100?	Many thanks for this remark, on an important subject that we completely neglected to highlight. Our reasoning is based on the paper "Abernethy, S., & Jackson, R. B. (2022). Global temperature goals should determine the time horizons for greenhouse gas emission metrics. Environmental Research Letters, 17(2), 024019." Where the authors highlight that the GWP period to be taken into account depends highly on the chosen climate-neutrality horizon. Reaching climate-neutrality by 2050 requires stabilisation of methane emissions by 2045, which means that we need to consider the period from now until that horizon: 22 years. This is much closer to GWP₂₀ than GWP₁₀₀. To clarify it in the text, we added the following subsection in the methods section in page 14, second paragraph: The choice of global warming potential period for methane leakage. The global warming potential of methane should be assessed considering the horizon of the climate target⁶⁰. For a climate-neutrality objective in 2050, the methane concentration in the atmosphere should be stabilised by 2045, and the period from the assessment date (2023) to this date (2045) should be considered as reference global warming potential assessment period, which is 22 years. Therefore, we chose GWP₂₀ as the reference for translation of methane emissions in CO_{2eq} terms.
Hydrogen also has a GWP20 of ~30 and GWP100 ~8 (see e.g., https://doi.org/10.1016/j.ijhydene.2022.11.219). Has this also been included, given that the paper results see nearly a quarter of final energy from hydrogen in 2050?	While we consider the main GHG emissions that are concentrating the atmosphere, we haven't studied the inclusion of hydrogen leakage in our study because of the lack of the empirical data. For methane leakage, we have the measurements via different sources that we highlight in the study, and for CO₂ emissions we know how the technologies emit CO₂. However, for quantifying the impact of hydrogen leakage, we need to find some widespread generalised hydrogen leakage data that is currently not available. Moreover, methane leakage could have been overcome (partially) historically, but during the development of the natural gas upstream activities, environmental concerns were not the priority of the industry. However, hydrogen value chain will be developed because of the environmental concerns, and

	therefore this new industry can in fact start correctly from the scratch. While we are not able to include it quantitatively to our study, we added this excellent point in a last paragraph to the discussion of the paper, in page 9, paragraph 4 as below: While we analyse the role of the two main GHGs of today (CO₂ and CH₄), in a future with nearly a fourth of final energy consumption satisfied by hydrogen, hydrogen leakage can also cause significant global warming effect (with 30 times higher potency than CO₂ over 20 years). In our analysis we use empirical data to assess the methane footprint of natural gas, and the technology data to assess the CO₂ emissions. Such a data for hydrogen leakage can be available once there is a widespread hydrogen value chain. Nevertheless, as this hydrogen value chain is in its early development phase, best practices regarding the minimisation of hydrogen leakage can be put in place by the potential hydrogen suppliers, transporters and consumers to not repeat the historical error of the oil & gas industry.
What is the reason for so much blue hydrogen in 2030? Is it cost driven or are their limits in the model for the scale-up rates of electrolysis?	While there are limits on the renewable capacity expansion (deployment rates), the reason for so much blue hydrogen in the model in 2030 is cost driven. The breakeven between blue and green hydrogen is not clear as there is a spread between the prices, and from 2035 onwards some of the green hydrogen becomes cost competitive compared to blue hydrogen. Moreover, the GHG-neutrality constraint is for 2050, while the lifetime of the hydrogen production technologies does rarely exceed 20 years. Therefore, the model chooses to install more blue hydrogen in the short term, as the 55% emission reduction constraint does not exclude blue hydrogen in this period, and it doesn't produce stranded assets for 2050.
I didn't understand how the capture rates are treated for blue hydrogen. The SI mentions a rate of 95%, which is much higher than current facilities achieve. Then the text says "were accounted for by assuming economic offsets". Why does unabated CO₂ not go into the model's CO₂ balance?	The 95% corresponds to the description of the HyPE model we believe. While we consider a CO₂+CH₄ neutrality for natural gas imported to Europe, for hydrogen imports, we don't include the upstream methane emissions, nor the residual CO₂ emissions in this balance. Therefore, the unabated GHG emissions from H₂ production outside Europe (in Hype) are not introduced into MIRET-EU's CO₂ balance but is accounted for as a higher H₂ import price. we introduced a CO₂ tax in Hype to account for these unabated

	emissions instead of transferring these emissions to MIRET-EU's CO₂ balance to better model the economic competition between the non-EU countries on the H₂ export market. We agree that the description of "climate neutrality" in the fundamental pillars of HyPE can be problematic, as we don't take indirect unabated emissions of low-carbon hydrogen production outside Europe in our GHG balance, and we only consider "economic offset" of it by addition of tax. Thus, we replaced "Climate neutrality of European energy imports" by "Low-carbon European energy imports" in the Supplementary Information 2 (HyPE).
Is net-zero emissions for CO₂ and CH₄ appropriate? Doesn't it depend on what is going on in agriculture and LULUCF sectors? It could be net-positive or net-negative depending (assuming an overall net-zero GHG target).	We agree that it can be economically beneficial to operate the energy system at either slightly above or below net-zero, depending on the assumptions with respect to LULUCF and agriculture as other solutions can be more, or less cost effective than the decarbonisation in the energy sector. As an example, "A Clean Planet for All"⁴ assumes net negative emissions through LULUCF of -236 Mt/year in 2050, which were decreased in the "Reference Scenario 2020"⁵ to around -270 Mt/year. However, the actual impact of LULUCF and forestry is difficult to estimate and depends as well on the actual trajectory in climate change (e.g., if the world is heading for 4 °C warming, emissions from LULUCF and agriculture might be different than in a 1.5 °C scenario. Hence, we focused on achieving net-zero only in the energy system. Based on other analyses, this can be seen as conservative due to net-negative emissions from LULUCF, but we believe that the energy system should not be relying on net-negative emissions in other sectors.
Are the costs of applying BAT technologies included in the total system costs? In the discussion it says "In contrast, the costs for achieving BAT deployment in the natural gas value chain are below 1% of the additional cost of the CEF scenario", so I wasn't sure.	We add the cost of the BAT technologies in an ex-post manner, by adding to the total cost (optimised objective variable of the model) once the optimisation is done. As natural gas is economically interesting, in case of less GHG footprint, it will reduce the cost of the energy system and this cost reduction is much more than the additional cost of the BAT adoption. But as we also have environmental constraints and limited geological carbon storage capacity, as its GHG footprint (methane emissions) increases, it will have

⁴ <https://eur-lex.europa.eu/legal-content/EN/TXT/?uri=CELEX:52018DC0773>

⁵ <https://op.europa.eu/en/publication-detail/-/publication/96c2ca82-e85e-11eb-93a8-01aa75ed71a1/language-en/format-PDF/source-219903975>

	less share in the energy supply (CEF scenario). This is also the base of the scenarios: we consider cost-optimal scenarios that are not optimised regarding the methane abatement development (exogenous methane abatement assumptions), and we compare the three scenarios. We excluded the explanation of energy system cost difference, but we added the following clarification in the Discussion and conclusion section, in the end of page 8 and beginning of page 9: The cost for methane abatement was not endogenously included in the model, and hence, may lead to an unaccounted price increase for natural gas in BAT compared to CEF. However, a back-of-envelope calculation of the cost of methane abatement based on IEA's methane tracker cost data shows that the cost for methane abatement is only a small fraction of the cost of natural gas. In fact, it corresponds to 0.4% of the cumulative costs for natural gas in the 30 years' perspective (below 1% of the overall energy system cost). Hence, not including the costs endogenously seems acceptable in combination with the general uncertainty in future natural gas prices.
Do downstream costs include leakage in distribution grids or end-use devices?	We introduced additional emission factors to natural gas consumption in MIRET-EU, that indeed take into account leakage on the downstream part of the natural gas value chain (transmission + distribution). These emission factors are used in the GHG emissions reduction constraints. However, no additional cost was added to the consumption of natural gas due to leakage.
From Figure 4 it was unclear what is going on with aviation and maritime fuels. The "feedstock for transport" seemed too small to cover international travel with e-fuels.	The aviation sector within Europe uses, depending on the scenario, 21-26 % e-fuels in 2050 while the remainder is covered by either biofuels or oil. Similarly, the maritime transport satisfies 60-61% of its demand via e-fuels resulting from hydrogen, with the remaining satisfied by oil, electricity and bioenergy. The fossil-based fuel consumption requires compensation through BECCS or DACCS. Neither the aviation sector nor the maritime transport sector from Europe to outside Europe is included in the analysis. We added the following precision in page 6, first paragraph, between lines 5 and 7: In the latter, hydrogen is mostly used directly in fuel cells to decarbonise heavy-duty vehicles (90% of heavy-duty vehicles' energy demand by 2050), and through the

	production of e-fuels for aviation (21 to 26% of its final energy demand by 2050) and bunkers (about 60% of the maritime transport final energy demand). We also added a footnote at the end of the sentence to clarify that these two sectors do not include inter-continental transport, and they do not reach climate neutrality by 2050 and they use offset capacities of BECCS and DACCS, as below: The analysis considers only the European maritime and aerial transport. Therefore, inter-continental bunkers and aviation is excluded from the analysis. Moreover, these two transport sub-sectors do not reach climate neutrality on their own by 2050, and they use the offset generated by BECCS and DACCS.
Figure 8: 1.4 GtCO₂/a sequestration is pretty high, compared e.g. to JRC-EU-TIMES scenarios and others. For a technology with a couple of MtCO₂/a at the moment. I'm not suggesting the authors change anything, it's just a comment.	We agree that our 1.4GtCO₂/year seems relatively optimistic compared to other scenarios. In most studies, the CO₂ storage potential is either assumed for the specific context of a study or alternatively used as a hard bound seemingly without investigating the actual potential. In this study, we aimed instead to base the total CO₂ storage level on a more rigorous analysis. The methodology for the calculation of this potential was developed as below, that is also added as a subsection in the Methods section in page 13, paragraph 3: The feasible level of annual CO₂ injection potential in the study has been estimated based on a survey of available knowledge about the potential for carbon storage in Europe, both in terms of total available geologic potential, which sets the overall frame for accumulated storage of European captured CO₂, and regarding the progressive increase in annual injection capacity, taking a sensible view on what is feasible in terms of drilling and sequestration in the absence of societal and regulatory barriers. In terms of total storage capacities in Europe, the United Kingdom⁶ and Norway⁷ hold the largest CO₂ storage potential by far with an offshore storage capacity of 70 Gt CO₂ each. The Dutch storage potential is assessed at 1.7 GtCO₂⁸. Ringrose and Meckel (2019) investigated

⁶ <https://www.bgs.ac.uk/geology-projects/carbon-capture-and-storage/>

⁷ <https://www.npd.no/en/facts/publications/co2-atlases/>

⁸ https://www.sintef.no/globalassets/project/elegancy/deliverables/elegancy_d5.2.3_ccs-in-the-netherlands.pdf

	the potential for scaling up CO₂ storage on the Norwegian continental shelf⁹. They estimated that offshore wells have an average injection rate capacity of 0.695 ± 0.222 Mt per year. Using historical data for well performance for the Norwegian North Sea, an estimated 2083 wells could be active by 2050 (using 2020 as initiation point for well development) in Norway. This corresponds approximately to an available injection rate of 1.4 Gt CO₂ per year by 2050 for Norway alone. In light of this information from projects and the literature and given the remaining uncertainty on future potential, a maximum of 1.0 GtCO₂ stored annually by 2040 and 1.4 GtCO₂ by 2050 – across Europe – has been determined as an adequate constraint for the study. We dropped the presentation of the results on the CO₂ storage, nevertheless, we added this precision (as written above) in the methods section.
please mention MIRET-EU only has 12 time slices per year. For HyPE, please be more concrete with "high spatial and temporal resolution" since this is relative.	We added in the methods section the temporal precision of both of the models. In page 10 paragraph 2 and lines 12 and 13 for MIRET-EU, we added the following precision: Spring, summer, fall, and winter are the four seasons considered by MIRET-EU, with each season broken down further into day, night, and peak resolution And for the HyPE model, we added the following precision in page 10, paragraph 3 and lines 1 to 3: HyPE is a hydrogen delivery chain optimisation model with high spatial (0.5°) and temporal (hourly time slices for at least one full year) resolution grounded on the literature on international hydrogen trade.
SI Figure 8: Very nice that you included country-specific WACC! You see it in difference of LCOH in Saudi Arabia versus Yemen.	Many thanks for your attention. Exactly, political risks are also reflected in our study via World Bank's country specific WACC values, based on its Ease of Doing Business scores.
For solar 170 MWp/km² is a density for the panels themselves. PV farms rarely reach above 70 MWp/km².	Many thanks for pointing this error out. There was a typing error. We assume 40% of the land covered with solar panels, and an average module efficiency of about 18%. Assuming an average direct irradiation of 950W/m², we find a power density of 68W/m² for a solar farm (equivalent to 170W/m² for panels) following Tröndle

⁹ <https://www.nature.com/articles/s41598-019-54363-z>

	(2020)'s assumptions¹⁰. We modified it in Supplementary Information 2 in page 3, paragraph 2, lines 8 to 10 as below: Power density describes the installable capacity over an area. We assume a 170W/m² of power density for PV modules and a 40% of farmland coverage by PV modules, leading to a 68W/m² of solar farm energy density
--	--

¹⁰ <https://journals.plos.org/plosone/article?id=10.1371/journal.pone.0236958>

REVIEWER COMMENTS

Reviewer #1 (Remarks to the Author):

Dear Authors,

Thank you for considering my comments and for your work in revising the article. After carefully reading through your response, the revised manuscript, and supporting materials, I am generally satisfied with your revisions and suggest that this article be published after minor revisions.

In my view, the article has benefited tremendously from the restructuring and the content is much more coherent and concise.

Here are my suggested revisions:

1. Improve the coherence between title, abstract and structure of the sections. Specific examples:

- The article's title focuses on methane leakages, yet four out of six findings in the abstract do not refer to methane at all. I strongly suggest the authors rewrite the abstract to align it better with the title.
- As a minor style suggestion, the authors should not use numeration in the abstract (this is, (1), (2), etc).
- The two first paragraphs in the introduction again do not talk about methane leakages, and lead with renewables, efficiency and hydrogen. I would encourage the authors to rewrite the introduction to lead with methane leakages earlier on.

2. Improve the accuracy and conciseness of the text. Specific examples:

- Lines 89-96. The authors make use of "both scenarios", "this scenario" in a confusing manner. Please, be more accurate and explicit even if the names of the scenarios have to be repeated.
- Why do you need figures 2 and 3? What is the point of showing primary and final energy? The use of both primary and final energy is confusing. For example, in lines 128 to 135, the authors talk about final energy consumption but refer the readers to figure 2. I would suggest the authors to choose one basis to show the results (final energy) and stick with it in figures and text. It is remarkable that fossil gas primary energy in the BAT scenario remains slightly below 30% while the final energy falls from 20% to nearly 0% between 2016 and 2050. All the more surprising because total final energy is lower in 2050 than in 2016, while total primary energy is larger in 2050 than in 2016. Therefore, the same share of primary energy in 2050 would correspond to a larger nominal amount of energy than in 2016. Can you

please explain this? Is it because fossil gas in figure 3 is contained in electricity and hydrogen final demands? I would encourage the authors to at least provide a footnote to avoid confusions to future readers.

- Although the overall focus of the article has improved greatly, the space and structuring of findings not immediately related to methane leakages is surprising. In lines 182-191, clearly the most relevant result related to the impact of methane leakages scenarios is hidden in the middle of the paragraph (lines 186-187, "the role of reformer-based ... depends on mitigation ... along the value chain."). This is the key contribution in this section. I suggest the authors revise the text to highlight and lead the structure with the most relevant findings to help readers grasp their contribution more easily.

- Perhaps more importantly, the discussion is also structured in a strange way and some statements are debatable. For example, in lines 213-215, "(1) the current ...". I do not think the results support the statement that the HP scenario is "insufficient for reducing methane emissions" for two reasons: (a) What do you mean by insufficient? The scenario meets the net-zero target ****by design****. So insufficient with respect to what? The authors could have discussed this in the next paragraph of the discussion but instead they jump to a topic unrelated with their specific findings. And (b) methane emissions are indeed reduced in HP with respect to CEF. Please consider these comments and improve the accuracy of the text.

- As said before, I would suggest structuring the discussion section differently. First, I like the two highlighted results in the first paragraph. However, I expected the authors to discuss the implications of these results (e.g., what does finding (1) mean for methane policies? What should policymakers and companies do about it?) in the following paragraphs 2 and 3 in the discussion. Current paragraph 2 on the increase of LNG is a new result, which should either be introduced earlier and here discussed the implications, mention in a brief manner, or not discussed (as it is of secondary importance, in my view). The current 3rd paragraph in the discussion is a welcome update that recognizes a limitation of the study. However, I would position it at the end of the discussion (as the authors have done with the paragraph on hydrogen leakages).

Other minor comments:

- There are recent studies that the authors could refer to when they mention the challenges to import hydrogen to Europe, such as related to infrastructure potential:

--Nunez-Jimenez and De Blasio 2022 <https://doi.org/10.1016/j.ijhydene.2022.08.170>

...and water availability:

--Pflugmann and De Blasio 2021 <https://www.ceeol.com/search/article-detail?id=877414>

--Tonelli et al. 2023 <https://doi.org/10.21203/rs.3.rs-2724691/v1>

- Lines 407-409: How does the timing of emission abatement options is represented in the model? Since those with no costs are achieved by 2030 and those with cost by 2035 and the model operates with 10-

years periods: Are both included in the period 2030-2040, or ones in the 2020-2030 and the others in the 2030-2040?

- In supplementary information 2 (about the HyPE model), in lines 49-51, you mention the availability of land and water. In lines 75-90, you explain how you limited land availability. But how did you limit water availability? I could not immediately find it, and it could be a major obstacle (if no desalination is considered, as in Nunez-Jimenez and De Blasio 2022) or increase costs and modify import routes (as discussed in IRENA 2022 <https://www.irena.org/publications/2022/Jul/Global-Hydrogen-Trade-Outlook>).

Reviewer #2 (Remarks to the Author):

The authors have thoroughly addressed my comments, so I recommend accepting the paper. It addresses an important topic and has interesting results for the literature.

The authors have mentioned that they do not address potential hydrogen leaks, which is good. This does give me pause for thought - its future role is smaller than today's methane, but it is a much smaller molecule and is more likely to escape from flanges, valves and pipelines.

Minor corrections:

Line 55: "different GHGs" - it's unclear what "different" refers to - different to CO₂ and CH₄? Consider replacing with "multiple".

Response to referees

The impact of methane leakage on the role of natural gas in the European energy transition

Reviewer #1

First, we wish to thank warmly the reviewer for the insightful comments raised in her/his report that led to the revision of our paper based on her/his recommendations. We appreciate the previous revision that brought significant coherence and quality to the second version of our paper, and we believe that the current comments are also fully in-line with improvement of the paper. Therefore, we took all of the reviewer's comments into account and we address them. New text inserted in the revised version of our manuscript is indicated in ***bold italic***. The manuscript and the supplementary information files are provided in two files: track-changes (with modification in **red**) and clean version.

Reviewer comment	Answer
Improve the coherence between title, abstract and structure of the sections.	We agree, and we modified the paper accordingly. The three specific examples are also addressed as mentioned by reviewer.
The article's title focuses on methane leakages, yet four out of six findings in the abstract do not refer to methane at all. I strongly suggest the authors rewrite the abstract to align it better with the title.	We modified the abstract, to make it more adapted to the methane leakage. Now we have 3 out of 5 results on methane leakage. We only highlight the key role of renewables and potential of hydrogen to replace natural gas partially in some end-uses.
As a minor style suggestion, the authors should not use numeration in the abstract (this is, (1), (2), etc).	We adopt the suggestion, and we deleted all the numberings in the abstract.
The two first paragraphs in the introduction again do not talk about methane leakages, and lead with renewables, efficiency and hydrogen. I would encourage the authors to rewrite the introduction to lead with methane leakages earlier on.	Many thanks for this suggestion, we fully agree and thanks to this suggestion, the introduction section makes the case for the whole paper and the sections' content. We modified the introduction section further, to bring methane leakage earlier to the explanation of the context. It is now in the first paragraph, right after the arguments on energy transition and the existing literature's solutions. We reduced, reformulated and put the paragraph on role of hydrogen to the last paragraph, to make the case for the reason why we discuss hydrogen in this study.
Improve the accuracy and conciseness of the text	We thank the reviewer for this comment, and the examples. We modified several parts of the text to increase the conciseness, and we addressed the specific examples one-by-one below.
Lines 89-96. The authors make use of "both scenarios", "this scenario" in a confusing manner. Please, be more accurate and explicit even if the names of the scenarios have to be repeated.	Thanks for this point, we modified the paragraph as below (in bold italic): The best available technologies (BAT) scenario assumes EFs that could be achieved by adopting BAT by a certain year. The assumed year of BAT deployment depends on the country. IEA methane abatement options and industry targets were applied to countries on a case-by-case basis. In all three scenarios the EF drops progressively until 2050. The methane EF in the

	BAT scenario is respectively 70% and 65% lower than the 2050 EF in the CEF and HP scenarios. The BAT scenario sees the sharpest decrease in methane's additional environmental burden by 2050. These results show that there is significant room for further methane emission reduction (BAT), more than what is envisaged under the existing policy framework (HP).
Why do you need figures 2 and 3? What is the point of showing primary and final energy? The use of both primary and final energy is confusing. For example, in lines 128 to 135, the authors talk about final energy consumption but refer the readers to figure 2. I would suggest the authors to choose one basis to show the results (final energy) and stick with it in figures and text. It is remarkable that fossil gas primary energy in the BAT scenario remains slightly below 30% while the final energy falls from 20% to nearly 0% between 2016 and 2050. All the more surprising because total final energy is lower in 2050 than in 2016, while total primary energy is larger in 2050 than in 2016. Therefore, the same share of primary energy in 2050 would correspond to a larger nominal amount of energy than in 2016. Can you please explain this? Is it because fossil gas in figure 3 is contained in electricity and hydrogen final demands? I would encourage the authors to at least provide a footnote to avoid confusions to future readers.	The reason why we present both is to show how primary energy sources that are currently used in the end-uses are transitioned. For instance, currently we use a lot of natural gas in the industries, but also buildings. In the future, while natural gas can still remain a primary energy option, it will be transformed to hydrogen and electricity and not consumed directly in the end-uses. This is also what brings the possibility of CCS in hydrogen and electricity production. Total primary energy in 2050 is not larger than in 2016, it's the contrary. The reason for the reduction over time is the shift from fossil energy sources to renewables, where the produced energy is directly used as electricity in different end-uses while fossil energy sources are associated with significant thermal energy losses compared to electricity (heat pumps are several times more efficient than boilers, electric vehicles are about 3 times more efficient than internal combustion engine vehicles, etc.). Total final energy consumption is lower thanks to shift to more efficient end-uses such as electrification in heating and electric vehicles that need less final energy. On top of that, we include European Commission's efficiency increase programmes in different industrial processes. Thanks for highlighting the issue with referring to Figure 2, in fact the reference for final energy consumption is to Figure 3 that we corrected. We added footnotes at the end of caption of each of the figures for further clarification. For Figure 2 we added the following footnote: Primary energy decreases by at least 100 Mtoe between 2016 and 2050. This decrease is mainly due to the massive replacement of fossil fuels by renewables, where the energy supply is already mostly in its final consumption form (for instance electricity for wind and solar power and hydroelectricity). In contrast, in a highly fossil-based energy system, the primary energy demand tends to be higher due to conversion losses of fossil energy sources to final end-uses (electricity, transport, low-temperature heating, etc.). For figure 3, we added the following footnote: Final energy demand experiences an 11% to 13% decrease between 2016 and 2050. In a growing economic environment with positive GDP (gross domestic product), the growth in the economic activities is expected to lead to higher final energy

	demand. In our analysis, while the final energy in the form it is consumed (for instance transport demand in tonne-kilometres and heating demand in the form of thermal energy demand - TWh_{th}) soars, the final energy carrier's demand quantity shrinks. This decrease is due to efficiency measures taken in the industrial processes by 2030, as well as the shift to more efficient final end-use energy carriers such as electricity and hydrogen. For instance, both electric vehicles and heat pumps for space heating are about two to three times more efficient than their combustion-based counterparts (internal combustion engine vehicles and boilers).
Although the overall focus of the article has improved greatly, the space and structuring of findings not immediately related to methane leakages is surprising. In lines 182-191, clearly the most relevant result related to the impact of methane leakages scenarios is hidden in the middle of the paragraph (lines 186-187, "the role of reformer-based ... depends on mitigation ... along the value chain."). This is the key contribution in this section. I suggest the authors revise the text to highlight and lead the structure with the most relevant findings to help readers grasp their contribution more easily.	We agree with the reviewer on putting methane emission subjects on priority. To do so, in the mentioned section, we merged paragraphs 2 and 3, and we brought "the role of reformer-based low-carbon hydrogen..." in the beginning of former paragraph " , which became as the following: "The hydrogen production mix generally consists of a combination of renewable hydrogen and low-carbon hydrogen produced in Europe, complemented by non-European imports. The fast deployment of reformers with CCUS enables a rapid launch of the hydrogen economy in all three scenarios, with more than 50% of hydrogen produced with them in 2030. In the long run, renewable hydrogen based on electrolysis takes over as the primary source of hydrogen in Europe, providing between 50 MtH₂ (BAT scenario) and 75 MtH₂ (CEF scenario). By then, the contribution of natural gas reformers largely depends on the mitigation of methane emissions along the natural gas value chain: in the BAT scenario, reformer-based low-carbon hydrogen reaches 25% of the total hydrogen production by 2050, while in the CEF scenario with the highest methane EF, European hydrogen is produced almost exclusively via electrolysis. In all three scenarios European clean hydrogen demand is not fully satisfied by domestic production and is complemented by imports (up to 25% of European hydrogen demand by 2050)."
Perhaps more importantly, the discussion is also structured in a strange way and some statements are debatable. For example, in lines 213-215, "(1) the current ...". I do not think the results support the statement that the HP scenario is "insufficient for reducing methane emissions" for two reasons: (a) What do you mean by insufficient? The scenario meets the net-zero target **by design**. So insufficient with respect to what? The authors could have discussed this in the next paragraph of the discussion but instead they jump to a topic unrelated with	We agree, and we modified the discussion section. The mentioned example for instance is now as below: "The methane abatement scenario analysis highlights two important results: (1) methane abatement, alongside with carbon capture and storage, as well as increased LNG imports are crucial for a continued role of natural gas with larger abatement (BAT compared to HP) resulting in a more pronounced role. In fact, (2) the current policy framework for methane emission reduction along the natural gas value chain (HP) entails very limited methane leakage reductions,

their specific findings. And (b) methane emissions are indeed reduced in HP with respect to CEF. Please consider these comments and improve the accuracy of the text.	ultimately degrading the role of natural gas in the climate-neutral European energy system.”
As said before, I would suggest structuring the discussion section differently. First, I like the two highlighted results in the first paragraph. However, I expected the authors to discuss the implications of these results (e.g., what does finding (1) mean for methane policies? What should policymakers and companies do about it?) in the following paragraphs 2 and 3 in the discussion. Current paragraph 2 on the increase of LNG is a new result, which should either be introduced earlier and here discussed the implications, mention in a brief manner, or not discussed (as it is of secondary importance, in my view). The current 3rd paragraph in the discussion is a welcome update that recognizes a limitation of the study. However, I would position it at the end of the discussion (as the authors have done with the paragraph on hydrogen leakages).	Many thanks for this remark. We restructured the discussion following your suggestion. Now we introduce the two aspects in different order (following your previous comment) and differently, making a conclusion that is supported by our study. Our current second paragraph is (based on the reviewer’s suggestion) explaining what policymakers and companies can do briefly (but concisely) for the methane emission reductions. Please note that the aim of this research is not to advise policymakers and Oil & Gas companies to keep natural gas in the energy mix but to show the impact of inaction at the side of methane abatement. Therefore, we rather remain neutral in our discussion: “The continuous role of natural gas in the future climate-neutral energy system requires significant determination both at the policy and industry sides. Our study shows that only CCS is not the sole requirement of continued role of natural gas. Even with large-scale CCS development, current methane leakage levels lead to a near phase-out of natural gas from the climate-neutral European energy mix (CEF). While historically neglected, deployment of best available methane abatement technologies is cost effective. Even with very low natural gas prices such as \$20/MMBtu, a methane abatement cost of \$140/tCH₄ accounts for no more than 1% of the natural gas price considering current emission factors. BAT rollout is also a very robust strategy for the oil and gas companies, that can guarantee a future for natural gas if combined with large-scale CCS deployment. The extra cost of these technologies remains negligible compared to natural gas production costs and carbon abatement options, while the future natural gas market in such a paradigm (BAT) can be nearly three times a future with no methane abatement (CEF). The difference between the results of BAT and CEF scenarios and their methane emissions calls for including methane leakage in GHG policies, either via inclusion in taxation or quota mechanisms. Moreover, to avoid extra-European carbon leakage, upstream methane emissions should be included in the European carbon border adjustment mechanism.” We also added an explanation on the increased LNG imports in the results section, where we discuss primary energy mix, as below: Paragraph 1 in page 4:

	“The European energy mix was highly fossil-dependent in 2016, with coal, oil and natural gas representing more than 1,100 Mtoe of primary energy demand (73%). By 2030, following the phase-out of Russian gas, the share of natural gas shrinks across all scenarios (Figure 2.a). By this date, the LNG import capacity in Europe is limited and only the BAT scenario sees an increase in LNG imports (about 40% higher than historical levels), partially replacing imports from Russia. Fossil energy sources represent less than 800 Mtoe of primary energy by this horizon. Although the chosen methane emission scenario has a strong influence on the role of natural gas and renewables in the energy mix in the long term (2050), oil and coal are set to dwindle: no matter the scenario, reaching net-zero by 2050 requires a near phase-out of these two energy sources.” At the same page, at the end of the second paragraph, we added the following sentences: “The only difference between the different scenarios being the assumed trend of methane emissions of natural gas, we found that higher footprints lead to higher substitution of natural gas by renewable energies. Natural gas represents only 9% of the primary energy demand in 2050 in the CEF scenario, where the footprint of natural gas is at the current high levels, while it represents as high as 26% of the primary energy mix in the BAT scenario where methane emissions are the most abated (Figure 2.b). Such a high share for natural gas in the primary energy mix means that about 70% of European natural gas consumption should be satisfied by LNG imports in the long run. Compared to the historical import levels, this amounts to two- (HP) to nearly four-fold (BAT) increase in European LNG imports.” We also agree with the reviewer’s comment on 3rd paragraph, and we put it at the end, just before the hydrogen leakage paragraph.
There are recent studies that the authors could refer to when they mention the challenges to import hydrogen to Europe, such as related to infrastructure potential: --Nunez-Jimenez and De Blasio 2022 https://doi.org/10.1016/j.ijhydene.2022.08.170 ...and water availability: --Pflugmann and De Blasio 2021 https://www.ceeol.com/search/article-detail?id=877414 --Tonelli et al. 2023 https://doi.org/10.21203/rs.3.rs-2724691/v1	We added the citation to the first paper at the end of the sentence below: “Although the amount of renewable energy needed for hydrogen exports from these countries to Europe is very limited compared to their land availability, ramping up the import capacities towards Europe can be challenging in practice.^{45”} About the scarcity of water and the associated costs, we already consider this issue in our modelling. We added an explanation in the SI 2(see the response to the last comment). We added citations to the second and third papers by addition of a sentence on the potential water-availability challenges, and how we actually handle this issue in our modelling:

	“Moreover, water consumption for renewable hydrogen production can limit the water availability in the exporting regions, that are already facing potable water scarcity issues^{46,47}. However, hydrogen production via electrolysis in HyPE uses water produced from seawater desalination only in the areas within a reasonable distance from the seas (see Supplementary Information 2).”
Lines 407-409: How does the timing of emission abatement options is represented in the model? Since those with no costs are achieved by 2030 and those with cost by 2035 and the model operates with 10-years periods: Are both included in the period 2030-2040, or ones in the 2020-2030 and the others in the 2030-2040?	The model incorporates the data coming from methane EF calculations. As the EF calculation analysis has a higher granularity than the MIRET-EU model, we consider the costless abatement for the year 2030, and the ones with cost are in 2040. We added the following footnote to the end of the mentioned paragraph: As the MIRET-EU model has 10-year simulation periods, it includes the abatement options with no net costs at the year 2030, and the ones with cost are included from 2040 onwards.
In supplementary information 2 (about the HyPE model), in lines 49-51, you mention the availability of land and water. In lines 75-90, you explain how you limited land availability. But how did you limit water availability? I could not immediately find it, and it could be a major obstacle (if no desalination is consider, as in Nunez-Jimenez and De Blasio 2022) or increase costs and modify import routes (as discussed in IRENA 2022 https://www.irena.org/publications/2022/Jul/Global-Hydrogen-Trade-Outlook).	In fact, we add the cost of water to the hydrogen produced. But we haven't mentioned it clearly in the Supplementary Information 2. We added the following explanations in Supplementary Information 2, at the end of page 4 and beginning of page 5 (last paragraph of page 4 and three first paragraphs of page 5): Water availability and competition with other uses is a major topic in particular in regions with resource scarcity as in some parts of Middle East, North Africa, Sub-Saharan Africa, Australia and Chile. Concerning the current study, Middle East and North Africa are the key clean hydrogen exporters to Europe. To internalise this issue, we follow a similar approach as that commented by IRENA in its global hydrogen trade outlook⁸. We assume that for acceptability reasons all water consumption of electrolyzers comes from seawater desalination. Accordingly, only sites within 300 km from the sea are considered and the associated costs of water supply are included in the LCOH calculations. Water desalination is already supplying about 95 million m³/day of water and producing 142 million m³/day of brine⁹. Curto et al.¹⁰ discuss the state of play of desalination technologies and affirms the next frontier for key commercial technologies such as reverse osmosis (RO), multi-stages flash desalination (MSF) and multi-effect distillation (MED), is to be powered by renewable sources. We base our water cost calculations on the technoeconomic figures reported by them for the different technologies considered in the model.

We estimate water cost by adopting an amortisation logic to desalination plants which includes capital and operational expenditures over its economic lifetime. We assume that the electricity used by desalination units comes from the power grid of the countries considered. Hence, our operational costs include costs of the electricity used, and the expenditures for offsetting the associated emissions of grid electricity. With the data of the electricity mix in each of the countries we estimate average electricity prices. The carbon intensity of the electricity supply leads to a carbon footprint of the desalinated water. We apply a carbon tax in line with the EU ETS, reaching €250/tCO₂ by 2050 for penalising for the associated emissions, and more importantly, to implement a level-playing field between countries with different electricity mixes (this is currently being discussed in the design of carbon border adjustment mechanisms). These values are compared to the water production cost estimated by the World Bank¹¹ and calibrated based on these values.

Reviewer #2

First, we wish to thank warmly the reviewer for the insightful comments raised in her/his first report that led to the revision of our paper based on her/his recommendations. Please find below our reactions to your final comment. New text inserted in the revised version of our manuscript is indicated in ***bold italic***. The manuscript and the supplementary information files are provided in two files: track-changes (with modification in red) and clean version.

Reviewer comment	Answer
The authors have mentioned that they do not address potential hydrogen leaks, which is good. This does give me pause for thought - its future role is smaller than today's methane, but it is a much smaller molecule and is more likely to escape from flanges, valves and pipelines.	We completely agree. And this is a very interesting research topic by itself that we must consider studying later. Many thanks for underlining this issue, that we noted in our further research agenda.
Line 55: "different GHGs" - it's unclear what "different" refers to - different to CO2 and CH4? Consider replacing with "multiple".	Many thanks for the suggestion. We modified the text accordingly to the phrase below: "While integrated assessment models (IAMs) can generally include multiple GHGs, they lack the detailed representation of the energy system and temporal and spatial precision, and they are generally based on top-down allocation of energy sources and carriers and they do not result from explicit optimisation."